# Learning with filopodia and spines: Complementary strong and weak competition lead to specialized, graded, and protected receptive fields

**Albert Albesa-González** ⬤ *, **Claudia Clopath** ⬤

Department of Bioengineering, Imperial College London, London, United Kingdom

* a.albesa-gonzalez22@imperial.ac.uk

## Abstract

Filopodia are thin synaptic protrusions that have been long known to play an important role in early development. Recently, they have been found to be more abundant in the adult cortex than previously thought, and more plastic than spines (button-shaped mature synapses). Inspired by these findings, we introduce a new model of synaptic plasticity that jointly describes learning of filopodia and spines. The model assumes that filopodia exhibit strongly competitive learning dynamics -similarly to additive spike-timing-dependent plasticity (STDP). At the same time it proposes that, if filopodia undergo sufficient potentiation, they consolidate into spines. Spines follow weakly competitive learning, classically associated with multiplicative, soft-bounded models of STDP. This makes spines more stable and sensitive to the fine structure of input correlations. We show that our learning rule has a selectivity comparable to additive STDP and captures input correlations as well as multiplicative models of STDP. We also show how it can protect previously formed memories and perform synaptic consolidation. Overall, our results can be seen as a phenomenological description of how filopodia and spines could cooperate to overcome the individual difficulties faced by strong and weak competition mechanisms.

## Author summary

Changes in the strength of synaptic connections between neurons are thought to be the basis of learning in biological and artificial networks. In animals, these changes can only depend on locally available signals, and are usually modeled with *learning rules*. Based on recent discoveries on *filopodia*, a special type of synaptic structure, we propose a new learning rule called Filopodium-Spine spike-timing-dependent-plasticity (FS-STDP). Our rule proposes that filopodia follow strongly-competitive STDP and spines (mature synapses) weakly-competitive STDP. We show that our model overcomes classic difficulties

**Data Availability Statement:** All python code, including code for generating the data and figures, is available from a public GitHub repository, https://

github.com/albesagonzalez/filopodium-spine-learning.

**Funding:** This work was supported by BBSRC BB/N013956/1 (C.C.), BB/N019008/1 (C.C.), Wellcome Trust 200790/Z/16/Z (C.C.), Simons Foundation 564408 (C.C.), and EPSRC EP/R035806/1 (C.C.). The funders had no role in study design, data collection and analysis, decision to publish, or preparation of the manuscript.

**Competing interests:** The authors have declared that no competing interests exist.

that these learning rules have separately, such as the absence of stability or specificity, and can be seen as a first stage of synaptic consolidation.

## Introduction

*Filopodia* are thin protrusions in dendrites [1] that have been long known to exist. Until recently, though, they were thought to play an important role only at developmental stages of brain formation [2]. However, filopodia have now been found to be more abundant than previously thought, as well as the structural substrate for silent synapses in the adult brain [3]. In fact, these structures make up to 30% of the dendritic protrusions. Because they lack AMPA channels, these synapses are effectively silent, meaning they cannot elicit a postsynaptic response unless the postsynaptic neuron is depolarized. Furthermore, filopodia are sensitive to plasticity induction protocols that are insufficient to potentiate spines (i.e., mature synapses that do contain AMPA channels). In particular, when applied the same spike-timing-dependent potentiation protocol, filopodia increased their synaptic efficacy but spines' remained the same. Finally, within minutes of being potentiated, the appearance of some filopodia started resembling that of a spine [3]. These findings beg the question: what are the underlying learning mechanisms of filopodia, and how do they coordinate with spines to facilitate cognitive functions? While there have been previous proposals of the distinct functional roles of filopodia and spines [4], experimental evidence that supports these has been scarce, and their relation to computational models of synaptic plasticity unexplored. The protocol used in [3] to change filopodia's strength is inspired by early studies of *spike-timing-dependent plasticity* (STDP) [5]. In STDP, changes in the weights' strength are a function of the difference in timing between pre- and postsynaptic neurons. Two prominent computational models of STDP are additive (add-STDP, see Table 1) [6, 7] and multiplicative (mlt-STDP and mlt/mlt-STDP, see Table 1) [8, 9]. add-STDP yields highly selective, bimodal receptive fields. However, it is intrinsically unstable, requires imposing hard-bounds on the weights, and is only able to capture the coarse structure of input correlations. On the other hand, multiplicative STDP creates a weight distribution that continuously matches the correlation structure of presynaptic input. These unimodal distributions are typically considered more realistic, since they better reproduce experimental results. However, the absence of synaptic specialization can hinder learning by mapping very different patterns to the same neuronal output activity. How these two different pictures can be reconciled has puzzled neuroscience modelling research for years [10]. One hypothesis is that the unimodal distributions found in experiments are in fact only the *observable* part of all synapses. These would be complemented with a big proportion of *silent synapses* [11, 12], which are not detectable via changes in postsynaptic potential, and would form another pool of effectively silent synapses. Whether these putative silent synapses were in fact present in the adult cortex remained, until now, largely unclear.

In this work, we present a computational model that explicitly distinguishes between filopodia and spines (Filopodium-Spine STDP, FS-STDP). We hypothesize that filopodia follow strongly competitive dynamics, implemented by approximating add-STDP, while spines learn in a soft-bounded, weakly competitive manner, associated with multiplicative models of STDP. As suggested by experiments [3], filopodia that undergo potentiation can be converted into spines, and our model assumes that the inverse is also possible. We turn to previous results of additive and multiplicative learning to predict the functional advantages of this type of combined learning, and use simulations to confirm our hypothesis. In particular, we show that FS-learning establishes a two-stage competition. The first stage is strongly competitive

**Table 1. Terms and definitions, spike-timing-dependent plasticity models.**

| Term | Description | Relevant Equation(s) |
|---|---|---|
| STDP | Spike-Timing-Dependent Plasticity: Plasticity mechanism in which the changes in synaptic strength depend on the precise timings of pre- and postsynaptic neurons' firings. Refers both to the experimental phenomenon and the learning rules describing it. | $\Delta w_{ij} = \begin{cases} f_+(w_{ij})e^{-(t_j - t_i)} & \text{if } t_j > t_i \\ -f_-(w_{ij})e^{-(t_i - t_j)} & \text{if } t_j \le t_i \end{cases}$ |
| add-STDP (additive-STDP) | STDP rule in which potentiation and depression are independent of the synaptic strength at update time | $f_+(w) = 1$ <br> $f_-(w) = \alpha$ |
| mlt-STDP (linear/ multiplicative-STDP) | STDP rule in which potentiation is independent of synaptic strength, but depression of synapse $i$ scales linearly with the synaptic weight $w_i$. While the label *mlt-STDP* is usually reserved for this model (see [19, 37]), it can also be called linear/multiplicative-STDP [10] to make explicit that only depression contains a multiplicative term. | $f_+(w) = 1$ <br> $f_-(w) = \alpha w$ |
| mlt/mlt-STDP (multiplicative/ multiplicative-STDP) | STDP rule in which potentiation and depression are both linearly dependent on $w_i$ [9]. It is important to note that both mlt-STDP and mlt/mlt-STDP share that a weak competition between synapses leads to unimodal distributions. For this reason, they are sometimes [13] simply called multiplicative STDP wihtout making an explicit distinction. | $f_+(w) = 1 - w$ <br> $f_-(w) = \alpha w$ |
| nlta-STDP (non-linear temporally asymmetric-STDP) | STDP rule that contains weight dependencies at potentiation and depression, but where this dependence is powered to a parameter $\mu$. $\mu$ is assumed to take values between 0 and 1. Note that for $\mu = 0$ one recovers the expression of add-STDP, and for $\mu = 1$ one recovers mlt/mlt-STDP. | $f_+(w) = (1 - w)^\mu$ <br> $f_-(w) = \alpha w^\mu$ |
| nlta*-STDP | nlta-STDP with a non-zero lower soft-bound $w_0$ | $f_+(w) = (1 - w)^\mu$ <br> $f_-(w) = \alpha\|w - w_0\|^\mu$ |
| FS-STDP (filopodium-spine-STDP) | STDP rule presented in this this study (together with with nlta*-STDP). It additionally makes $\mu$ dependent on $w$ | $f_+(w) = (1 - w)^\mu$ <br> $f_-(w) = \alpha\|w - w_0\|^\mu$ <br> $\tau_\mu \frac{d\mu}{dt} = -\left(\mu - \frac{w+a}{q}\right)$ |

and classifies synapses in a bimodal fashion (as does add-STDP) depending on presynaptic correlations. The second stage, governed by soft-bounded dynamics, continuously represents the input correlations in the spines that emerge from the first stage. We also show that the soft-bounded dynamics of mature spines shield them from becoming filopodia again. This makes the gross structure of the existing receptive field resistant to new correlated input, thus protecting previously formed memories from being erased instantaneously when environmental statistics change.

# Results

## FS-learning induces strong competition between filopodia and weak competition between spines

**Model assumptions and implementation.** Our model considers the existence of *filopodia* (Fig 1A, in grey), which can dynamically evolve through learning into *spines* (in purple). We

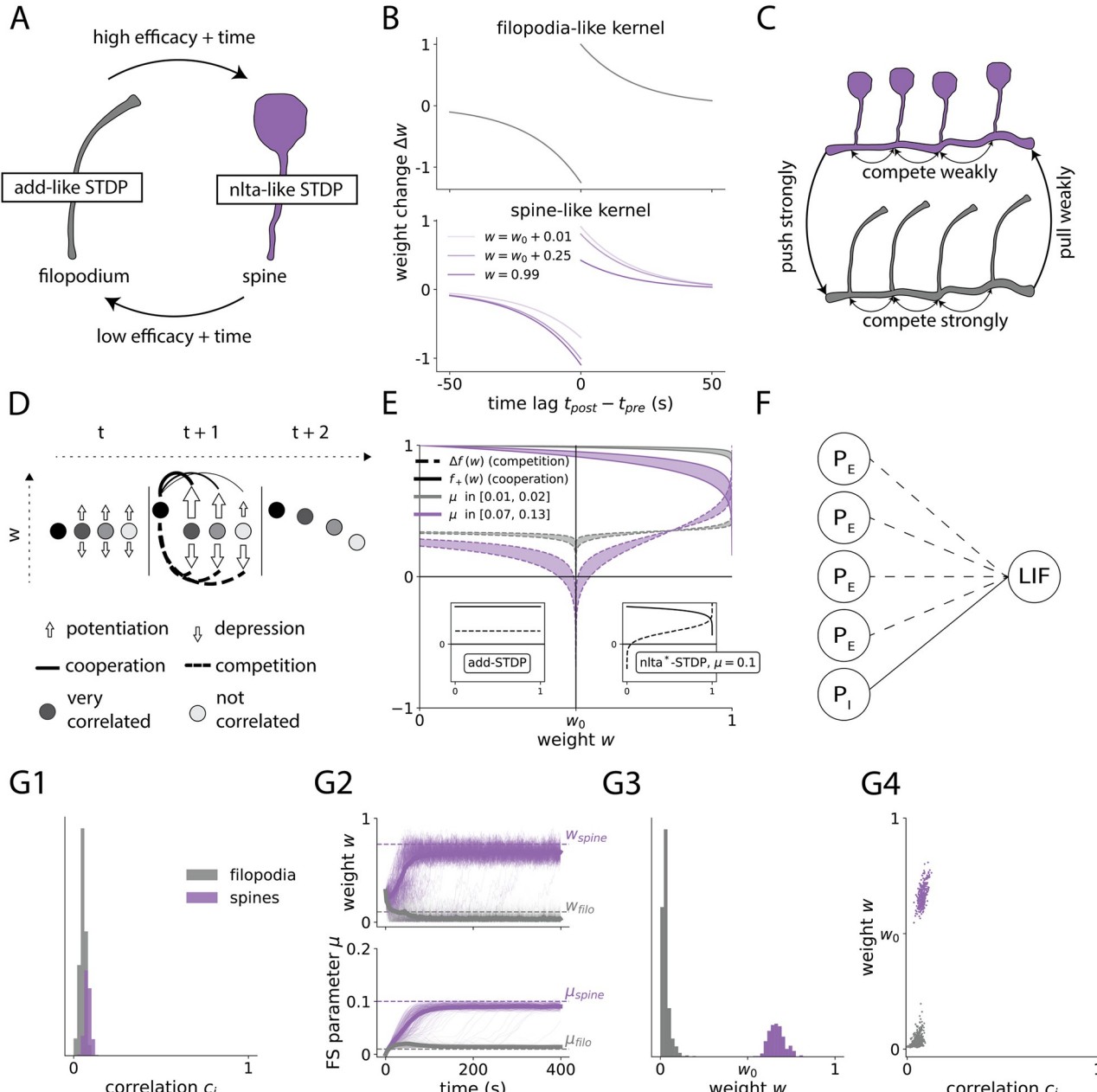

**Fig 1. Filopodia compete strongly to become spines, and spines compete weakly to represent input correlation. A**: Filopodium (grey)—Spine (purple) scheme. Filopodia can become spines if their efficacy is increased for a long enough time (observed experimentally in [3]). Similarly, spines can become filopodia if their weight is consistently low (our model). **B**: Learning for filopodia (Top) and spines (Bottom). Filopodium-like dynamics effectively lead to a strong competition, and spine-like dynamics to a weak competition. **C**: Diagram indicating the push-pull forces between the different types of synapses in our model. Spines are strong weights and compete weakly between them. Filopodia are silent synapses that compete strongly to become spines. While the push that spines exert over filopodia is strong, the pull that filopodia exert over spines is weak (due to the lower-soft bound $w_0$ in the dynamics of spines). **D**: Diagram of competition (dashed lines) and cooperation (solid lines). When a synapse increases its efficacy (left circle from $t$ to $t+1$), it has two opposite effects on the rest of the synapses. It equally increases depression (competition) and it also increases the potentiation of those it is correlated with (cooperation). **E**: Competition (dashed lines) and cooperation (solid lines) factors, as a function of weight $w$, for a fixed $\mu$. The areas are bounded by the curves corresponding to the $\mu$ range of typical filopodia (weight from 0 to 0.1) and spines (weight from 0.5 to 1). Small subpanels on the left and right show, respectively, how the same plot would look for add-STDP and nlta*-STDP. The shape of competition and cooperation factors is very similar to additive learning for weights below $w_0$, but resembles soft-bounded learning for efficacies above this value. For the exact expressions plotted see Table 6. **F**: Diagram of network architecture. Our model consists of one postsynaptic LIF neuron that receives many Poisson realizations of presynaptic input, which can be either excitatory ($P_E$) or inhibitory ($P_I$). Only excitatory synapses are plastic. Excitatory presynaptic neurons are temporally correlated via a *correlation structure* $\vec{c}$. The spike times of presynaptic neurons $i$ and $j$ will be correlated only if both $c_i$ and $c_j$ are sufficiently high ($C_{ij}^+ \sim \sqrt{c_i c_j}$). **G**: Example of FS-learning imprinting a pattern with correlation structure $\vec{c}$. **G1**: Distribution of correlation

strength $c$, sampled from a Gaussian distribution with mean 0.3 and standard deviation 0.1 (lower values clipped at 0) and then normalized to have $c_{\text{tot}} \equiv \sum c_i = 60$. Distributions for synapses that become filopodia (grey) and spines (purple). The label *filopodia* is assigned to synapses with a mean (over 10 seconds) weight smaller than $w_0$. Synapses with a value equal or higher than $w_0$ are labeled as *spines*. **G2**: Trajectories of synapses and $\mu$ under FS-STDP and a correlation structure as in Fig 1G1. **G3**: Distribution of weights after learning. **G4**: Scatter plot of final weight $w_i$ as a function of $c_i$.

hypothesize that filopodia, which are more sensitive to potentiation protocols, compete strongly to become spines, which is implemented by making them approximate additive STDP. In add-STDP, changes in weights depend exponentially on the differences of spike times between the pre- and postsynaptic neuron. Furthermore, these changes are independent of the synaptic state, so they are not affected by the current strength of the weight (Fig 1B, Top). As has also been found in experiments [3], we assume that if a filopodium is potentiated for a long enough time, it can be converted into a spine (Fig 1A). We propose that spines, in contrast, have soft-bounds that make weight changes dependent on the synaptic state (Fig 1B, Bottom). These soft-bounds, which are associated with models that induce a weak competition, can be found in mlt-STDP [8] (only for depression), mlt/mlt-STDP [9] and nlta-STDP [13] (for certain parameter regimes; for more details on the nature of these models see Table 1). All these learning rules have in common a higher degree of stability, and have been related to experimental results in spines [14, 15].

To describe our model mathematically, we make use of a modified version of *nonlinear temporally asymmetric* (nlta)-STDP [13]:

$$\frac{dw_i(t)}{dt} = \lambda(1 - w_i(t))^{\mu} z_i(t) S_{\text{post}}(t) \quad - \quad \lambda\alpha w_i(t)^{\mu} z_{\text{post}}(t) S_i(t) \tag{1}$$

where $w_i$ is the synaptic efficacy of synapse $i$, $\lambda$ the learning rate, and $z_i$, and $z_{\text{post}}$ the pre- and postsynaptic traces, and $S_i$ and $S_{post}$ the pre- and postsynaptic spike trains (see Methods for further definition of these variables). $\alpha$ indicates the imbalance between potentiation and depression for an equivalent spike time difference. In its original form (Eq (1)), nlta-STDP incorporates a parameter $\mu$ such that, depending on its value, it induces a stronger or weaker competition between synapses. For example, in the presence of a highly correlated subgroup of synapses, if $\mu$ is very small, the steady-state distribution of synaptic weights is bimodal, but as $\mu$ increases, the modes of the distribution get closer and closer, resembling what one obtains with mlt-STDP or mlt/mlt-STDP. In particular, for $\mu = 0$ one recovers exactly add-STDP, and for $\mu = 1$, mlt/mlt-STDP. As our model assumes that filopodia follow additive STDP, but spines have soft-bounded weight updates, a simple implementation could be that filopodia follow nlta-STDP with $\mu = 0$, and spines have a $\mu$ value that falls within the multiplicative range. However, instead of imposing this in a rule-based manner, we make parameter $\mu_i$ ($i$th synapse) low-pass filter the synaptic efficacy of that synapse $w_i$

$$\tau_{\mu} \frac{d\mu_i(t)}{dt} = -\left(\mu_i(t) - \frac{w_i(t) + a}{q}\right) \tag{2}$$

with $a$ and $q$ model parameters (which have implications in the synaptic consolidation aspect of this learning rule, see last section in Results) and $\tau_{\mu}$ is the time constant associated to the transformation of filopodia into spines. One can see how for a synaptic efficacy of 0, $\mu_i$ converges to $a/q \approx \mu_{\text{filo}} << 1$ (Eqs 2 and 9), so this fulfills the model assumption that silent synapses follow additive STDP. Then, as $w_i$ increases, $\mu_i$ also increases, leading to a degree of softness in the bounds that is controlled by $a$ and $q$. For better interpretability, instead of directly fixing $a$ and $q$, we define two model parameters $\mu_{\text{filo}}$ and $\mu_{\text{spine}}$, and solve Eq (2) for $a$ and $q$ under equilibrium for typical synaptic values of filopodia and spines (Eq 9 in Methods

and Fig 1G2). This way, $\mu_{\text{filo}}$ and $\mu_{\text{spine}}$ dictate what will approximately be the $\mu$ value (and thus level of competition) associated to filopodia and spines (respectively). Our choice of $\mu$ dynamics is also related to a potential connection with physiology. If, as seen experimentally, filopodia learn differently than spines, and if the transformation of one to the other is influenced by an increase in synaptic efficacy (unsilencing), it is very likely that this is governed by some physiological signal that takes time to be processed and integrated. Hence, we use a slow time constant that decouples the synaptic dynamics (happening in the order of milliseconds) and the evolution of filopodia into spines, which was reported to take place within minutes. We use a slightly quicker ($\tau_\mu = 20$s) time constant to speed-up simulations, which is still considerably slower than synaptic plasticity. Imposing slow dynamics in $\mu$ also has a functional motivation, as it makes learning more stable and less prone to oscillatory behaviours, as well as truly dependent on the long-term input statistics rather than instantaneous synaptic states. One last modification of nlta-STDP as originally proposed is in the weight dependence for depression, which takes the form of

$$\Delta w_i^{\text{depression}} \propto |w_i - w_0|^{\mu_i(t)} \tag{3}$$

such that the lower bound at 0 that soft-bounded models traditionally incorporate is generalized to an arbitrary synaptic efficacy $w_0$. This is crucial for the consistency of the model, as it maintains spines in an efficacy range such that their $\mu$ leads to a weak competition. Then, in turn, this leads to a synaptic distribution within that range. The implications of this form of depression for filopodia are not substantial, in the sense that their $\mu$ value is so small that depression is constant irrespectively of the synaptic strength. Instead, this bound is related to the consolidation aspect of the model, so that once a synapse has consolidated into a spine it is hard to be depressed beyond $w_0$, due to the existence of this lower bound (the synapse is protected). In view of the study that eminently inspires this computational work [3], one can think of $w_0$ as imposing a minimum order of magnitude in the synaptic efficacy of spines, which in that case would be around 0.1mV. This would explain why there is not a continuum of efficacies between zero and 0.1mV, as that region would be reserved to filopodia and would be intrinsically unstable (leads to either reaching the $w_0$ threshold and eventually become a spine or becoming silent again). Altogether, this gives rise to the final expressions (Eqs (4) and (2)) of our learning rule (FS-STDP), which we will investigate throughout this study:

$$\frac{dw_i(t)}{dt} = \lambda \overbrace{(w_0^+ - w_i(t))^{\mu_i(t)}}^{\text{nlta}^*} \overbrace{z_i(t)S_{\text{post}}(t)}^{\text{"vanilla" STDP}} - \lambda \overbrace{\alpha}^{\text{pot./dep. imbalance}} \overbrace{|w_i(t) - w_0^-|^{\mu_i(t)}}^{\text{nlta}^*} \overbrace{z_{\text{post}}(t)S_i(t)}^{\text{"vanilla" STDP}} \tag{4}$$

$$\tau_\mu \frac{d\mu_i(t)}{dt} = -\left(\mu_i(t) - \frac{w_i(t) + a}{q}\right) \tag{2}$$

nlta* refers to standard nlta-STDP with an arbitrary lower and upper soft-bound, or equivalently, FS-STDP with a fixed value of $\mu$ for all synapses.

**Competition profile of FS-STDP.** An interesting behaviour that collectively arises from plastic synapses in a feed-forward network is that of *selectivity*. For example, if presynaptic activity can be found in two different states when a stimulus A or a stimulus B is present in the environment, it is useful for the activity of postsynaptic neurons to convey information about the presence of either A or B. This can be achieved with a mix of *competition* and *cooperation* between synapses. One way of implementing competition is to have a mean synaptic depression proportional the total synaptic efficacy. This means that the higher the strength of a group of synapses is, the more the rest is depressed (Fig 1D, dashed lines). The notion of cooperation,

instead, refers to a scenario in which the average potentiation a synapse receives is proportional to the efficacy of other synapses. However, while competition is independent of the correlation in the activity between presynaptic neurons, cooperation is correlation-dependent, so when a synapse is potentiated, it favors the potentiation of other synapses, but only those its presynaptic activity is correlated with (Fig 1D). These notions of competition and cooperation can be related to the exact equations of an STDP model via a mean-field analysis of the learning rule (see Methods or [13]). This gives an approximation to the average instantaneous potentiation and depression $\dot{w}_i$ that a synapse $i$ receives:

$$\dot{w}_i = \overbrace{\frac{\lambda \tau_{\text{STDP}} r_{\text{pre}}^2}{N_{\text{pre}}}}^{(1)\ \text{effective learning rate}} \left[ \overbrace{- \underbrace{\Delta f(w_i)}_{\text{competition factor}} \sum_j w_j}^{(2)\ \text{correlation-independent interaction}} + \overbrace{\underbrace{f_+(w_i)}_{\text{cooperation factor}} \sum_j C_{ij}^+ w_j}^{(3)\ \text{correlation-dependent interaction}} \right] \quad (5)$$

One can break down this weight update (Eq (5)) into: (1) An *effective learning rate* that determines the overall rate of change in synaptic efficacy, which depends on the actual learning rate $\lambda$, the STDP time constant $\tau_{\text{STDP}}$, the presynaptic firing rate $r_{\text{pre}}$ and the total number of synapses $N_{\text{pre}}$. (2) A correlation-independent interaction, which depends on the difference between potentiation and depression expressions in the learning rule ($\Delta f(w_i) \equiv f_-(w_i) - f_+(w_i)$). These are the amplitudes of the negative and positive curves (respectively) in Fig 1B (also see Methods and Table 2). It is called an interaction because it also depends on the rest of the synapses via $\sum_j w_j$. Note how, for a positive $\Delta f(w_i)$, this interaction leads to the type of competition described above, as all synapses receive greater average depression if the total synaptic efficacy increases. For this reason, $\Delta f(w_i)$ is called the *competition factor* (Table 3). (3) A correlation-dependent interaction, which depends on $f_+(w_i)$ (called the *cooperation factor*, Table 3). Again, this relates to the previous notion of cooperation, as an increase in a synapse $j$ will increase the overall potentiation of $w_i$ only if the correlation between the corresponding presynaptic activity $C_{ij}^+$ is high. Given that this second interaction cannot have a depressing effect, if the competition factor $\Delta f(w_i)$ was negative, that would lead to an intrinsic instability of the synapse, as both the correlation-independent and correlation-dependent terms would be positive, so under equilibrium it is safe to assume that $w_i$ leads to a positive $\Delta f(w_i)$. It should be noted how Eq (5) is a mean-field approximation of STDP, but the notion of competition can also be intuitively extracted from its exact form (see Relevant Equations for STDP in Table 1). If one

**Table 2. Terms and definitions, mean-field dynamics.**

| Term | Description | Relevant Equation(s) |
|---|---|---|
| Strongly competitive learning (or dynamics) | When a pool of synapses is split into distributions around two distant modes, we say that the competition between them is *strong*. | |
| Weakly competitive learning (or dynamics) | When a group of synapses form a unimodal distribution that continuously represents input correlations within their input, we say that they are following multiplicative learning (or dynamics, and that the competition is *weak*). Because this distributions can be obtained with soft-bounded rules of STDP, they can also be called soft-bounded. | |
| Mean-field dynamics | Expression describing the mean weight change experienced by a synapse in a Poisson-linear model. This is an approximation of the actual trajectories followed in a Leaky Integrate-and-Fire (LIF) model, but gives rise to expressions (see below) that qualitatively describe the behaviour of the learning rule. | $\dot{w}_i = \frac{\lambda \tau_{\text{STDP}} r_{\text{pre}}^2}{N_{\text{pre}}} \left[ -\Delta f(w_i) \sum_j w_j + f_+(w_i) \sum_j w_j C_{ij}^+ \right]$ |
| Correlation-dependent (mean-field) interaction | This term is called correlation-dependent because the magnitude of its effect over a synapse $i$ depends on how its input is correlated with that of other synapses $j$ (via $C_{ij}^+$). | $f_+(w_i) \sum_j w_j C_{ij}^+$ |
| Correlation-independent (mean-field) interaction | This term, has a magnitude proportional to $\sum w_j$, so is the same for all synapses $w_i$ irrespectively of the input correlation structure. | $-\Delta f(w_i) \sum_j w_j$ |

**Table 3. Terms and definitions, competition and cooperation factors.**

| Term | Description | Relevant Equation(s) |
|------|-------------|----------------------|
| Cooperation factor | Within the correlation-dependent interaction, $f_+(w_i)$ is fixed by the learning rule, and it describes how the amplitude of synaptic potentiation depends on its specific state $w_i$. Because the greater the correlation-dependent term is the more a synapse is potentiated, and the more other correlated synapses also are, this is called the cooperation factor. This means that when a synapse increases its value, it favors the potentiation of other synapses with which it is correlated, which in turn can have the same effect back. This leads to a positive-feedback loop between correlated subgroups, effectively making correlated synapses *cooperate* to increase the overall subgroup synaptic efficacy. | $f_+(w)$ |
| Competition factor | In the correlation-independent interaction, also appears a factor ($\Delta f(w_i) \equiv f_-(w_i) - f_+(w_i)$) fixed by the learning rule. Because of the negative sign in front of the correlation-independent interaction, the sign of $\Delta f(w_i)$ will dictate if this interaction leads to potentiation or depression. In the case $\Delta f(w_i) > 0$. This interaction becomes depressing and proportional to the total synaptic efficacy ($\Sigma\, w_j$). This means that, when a synapse increases its weight, the rest (and itself) effectively experience a slightly more depressing field. This leads to a *competition* between synapses that try each to allocate for themselves a fraction of the total synaptic efficacy available. While in practice this term can sometimes be positive (Fig 1E), one can see how that only happens for a specific weight interval. If one synapse was in that interval, it would be intrinsically unstable, as both correlation-dependent and independent mean-field interactions become positive. No synapse can be found in this region under equilibrium, and it is safe to say that, effectively, this term results in competition. | $\Delta f(w_i) = f_-(w) - f_+(w)$ |

synapse increases its efficacy, it decreases the average spike-time difference between its pre- and postsynaptic neurons (and increases its own potentiation). However, it also increases the postsynaptic firing rate, which leads to an overall increase of depression for all synapses (assuming depression has a higher amplitude than potentiation). This influence of the total synaptic efficacy on the average depression experienced by all synapses is what the correlation-independent interaction (Eq (5)) represents. On the other hand, when another presynaptic neuron is temporally correlated with that of a synapse that has been potentiated, the difference in spike times will also (indirectly) decrease. This can compensate the extra depression, and allows the two synapses to cooperate in mutually increasing their weights. This is the effect described by the correlation-dependent interaction (Eq (5)).

Under this paradigm, the behaviour of a learning rule can be predicted by its *competition profile* (Fig 1E), which is simply the curves of both $\Delta f$ and $f_+$ as a function of the synaptic strength. add-STDP is said to induce a *strong competition*, because synapses are split between *winners* and *loosers*, thus following a bimodal distribution. Multiplicative models, like mlt-STDP or mlt/mlt-STDP, induce instead a *weak competition*, where more correlated synapses have bigger synaptic efficacy, but the distribution is rather unimodal. nlta-STDP can have different competition profiles depending on the chosen value of $\mu$, which is why it can lead to both strong and weak competition. In this context, one wonders what is the competition profile of FS-STDP, and how it compares to that of other models of STDP. The answer is that it depends on the synapse you are looking at, and whether it has become a filopodia or a spine. For the $\mu$ corresponding to typical synaptic efficacies of filopodia (which usually range from 0 to 0.1), the competition profile observed (grey areas in Fig 1E) are virtually independent of the weight $w$, being almost exact to add-STDP competition profile (left subpanel). Similarly, we find that for $\mu$ corresponding to a synaptic range of $w_0$ to 1 (range of spines), the competition

profile from $w_0$ to 1 is very similar to nlta-STDP with $\mu = 0.1$ (right subpanel). For this reason, for timescales where $\mu$ can be considered constant, changes of filopodia follow strongly competitive dynamics (irrespective of their instantaneous synaptic efficacy), and the same applies for spines and a weak competition. The study of the competition profile of FS-STDP confirms that this particular implementation is consistent with the assumption that filopodia's dynamics are approximately additive and strongly competitive, and spines following bounded dynamics, associated with a weaker competition. Furthermore, we would like to point out that while it is common to refer to models of STDP as strongly or weakly competitive, competition happens at the synapse pair level. In this sense, how filopodia and spines compete (between these two groups) is actually asymmetric. Because filopodia approximate add-STDP, they receive a high pushing force from spines, as the latter result in a high total synaptic efficacy. However, filopodia that could potentially become spines play with a disadvantage, which comes from the lower soft bound of spines. For this reason, the pull force that a filopodia could produce by increasing its synaptic efficacy is much weaker (Fig 1C) than it would if spines followed additive dynamics. This becomes an important result for FS-STDP as a model of synaptic consolidation, which is studied further below.

**An example of FS-STDP *in action*.** We now test with simulations the evolution of synapses following FS learning, connected to a postsynaptic neuron (Fig 1F), and in the presence of input correlations (Fig 1G1). We use a conductance-based Leaky Integrate-and-Fire (LIF) neuron that contains 1000 excitatory and 200 inhibitory presynaptic inputs, each modelled as a Poisson process (Fig 1F). Excitatory connections are plastic and follow FS-learning (Eqs (4) and (2)). Presynaptic inputs are not independent from one another, but instead have a temporal correlation structure $\vec{c} = \{c_i\}$ that determines how correlated the spike times of neuron $i$ are to the rest of the presynaptic pool (Fig 1G1). To do this, we generate a reference spike train, and then $c_i$ indicates how similar are the spike times of presynaptic neuron $i$ with that reference. Indirectly, this makes two neurons $i, j$ that are very correlated with the reference spike train also very correlated between them, inducing cross-correlations of the order $C_{ij}^+ \sim \sqrt{c_i c_j}$ (Methods, Eq (23)). All presynaptic neurons have the same temporally averaged firing rate, denoted by $r_{\text{pre}}$. In this initial example, we use a correlation structure sampled from a Gaussian distribution (Fig 1G1). Starting with a homogeneous synaptic state ($w_i = 0.3$ to avoid quiescence), if $\mu_i$ is set to 0 as initial condition (all synapses are filopodia), the mean field analysis predicts that filopodia will compete strongly, as in add-STDP, such that only a fraction of them (those more correlated) will increase their synaptic efficacy, pushing the rest of the synaptic pool to remain silent. Furthermore, after that first stage of competition is completed, winning synapses will increase their $\mu$, and start a competing weakly between them, distributing themselves around a single mode. As predicted (Fig 1G2, 1G3 and 1G4), due to the first stage of the competition, FS learning results in two qualitatively distinct groups, filopodia ($w < w_0$) and spines ($w \geq w_0$). These two groups have very different averages compared to their variance (Fig 1G3). In addition, as also predicted by the second stage of the competition, weights $w_i$ depend on the correlation value $c_i$ of each synapse (Fig 1G4). Simulations support our two-stage competition proposal, which predicts an initial bimodal classification (first stage) and a continuous representation of the correlation strength in the synaptic efficacy (second stage). Furthermore, we investigated whether the mean-field dynamics of filopodia approximate those of add-STDP and (and spines those of nlta*-STDP, which is nlta-STDP including the same lower bound $w_0$, see Methods or Table 1). We computed, at every timestep, the correlation-dependent and correlation-independent interactions (Table 2), as well as the competition and cooperation factors (Table 3). We did this for the correlation structure presented in this section (S1 Fig), a squared pulse (Methods and S2 Fig) and a von Mises structure (Methods and S3

Fig), which will be used throughout the rest of the study. Our results indicate that the competition and cooperation factors, as well as the mean-field interactions, of filopodia and of spines are very similar to those of an equivalent setup with add-STDP and nlta*-STDP (respectively).

## FS-learning represents input correlations better than add-STDP, and is more selective than mlt-STDP

Having seen that FS-learning inherits properties of both additive and multiplicative learning, now we quantify how the receptive fields obtained via filopodia and spines compare to pure add- and mlt-STDP. We choose von Mises shaped input correlations (Fig 2A) for our simulations for two reasons: First, the experimental study that motivates our research [3] is based on visual cortex cells. Second, this choice gives input patterns a rich non-binary correlation structure, which allows us to test the correlation representational power. In this sense, using simpler correlation structures like squared pulses does not allow a quantification of the learning rule sensitivity to input correlations (see S4 and S5 Figs). We stimulate our neuron for 200 seconds and then extract two measures: (i) the Pearson correlation $r$ between the developed synaptic efficacies and the correlation strength of their corresponding presynaptic activity (Fig 2B and Methods), and (ii) the Discrimination Index $DI$ that results from averaging the neuron response to different correlation patterns (Fig 2C and Methods). We use $r$ as a proxy to how well the formed receptive fields (RFs) represent input correlations, and $DI$ to quantify the ability of our neuron to discriminate different inputs. We start by noting the differences seen in both the weights trajectories and developed receptive fields (Fig 2D and 2E, respectively). Both FS and add learning result in very similar initial trajectories, which are associated with the first (strong) stage of the competition (Fig 2D1 and 2D2). In terms of the RFs formed, this initial strong competition makes both FS-STDP and add-STDP present a group of synapses with 0 efficacy along with a subgroup with significant synaptic strength (Fig 2E1 and 2E2). This is not the case of mlt-STDP, where all of the synapses are located around a single mean (Fig 2E3). If one focuses on non-zero synapses, however, then FS-STDP is more similar to mlt-STDP, as both learning rules continuously match the input correlation structure. In contrast, add-STDP erases this information and yields a binary RF resulting from classifying presynaptic input as correlated enough or not. We quantify the differences in discriminability (as measured by DI) and input representational capacity (as measured by $r$) of FS-learning across different values of potentiation-depression imbalance (parameterized with $\alpha$), and across different values of correlation strength (given by $c_{\text{tot}}$). These two terms control the ratio between the cooperation and competition factors, which strongly affects the formation of the RFs (see Eq (29) in Methods). We obtain $r$ and $DI$ for each combination of these two parameters, as well as for every learning rule (Fig 3A and 3B). $r$ measures the strength of the linear relation between a synapse $w_i$ and the correlation strength ($c_i$) of its corresponding presynaptic neuron with the rest of the pool. Assuming no higher order relationships exist between the two, we take this metric to indicate how well the input correlations are imprinted into the weight structure, with $r$ ranging from 0 (not well represented) to 1 (well represented). In general, there is a high degree of correlation in FS-learning (Fig 3A1). Due to the second stage of the competition, the input structure is much better imprinted in FS-STDP than in additive learning (Fig 3A2), reflected by higher $r$ values for FS-STDP. In terms of discriminability, FS-STDP performs similarly to add-STDP (Fig 3B2, $FS - add$). The lack of synaptic specialization induces a discriminatory capacity of mlt-STDP near zero. Our results propose a biologically plausible explanation of how highly specialized RFs that continuously represent the input correlations can emerge via the strong competition of filopodia and the weak competition of spines. In addition, it explains how unimodal yet selective distributions could arise in the experimentally observed synaptic

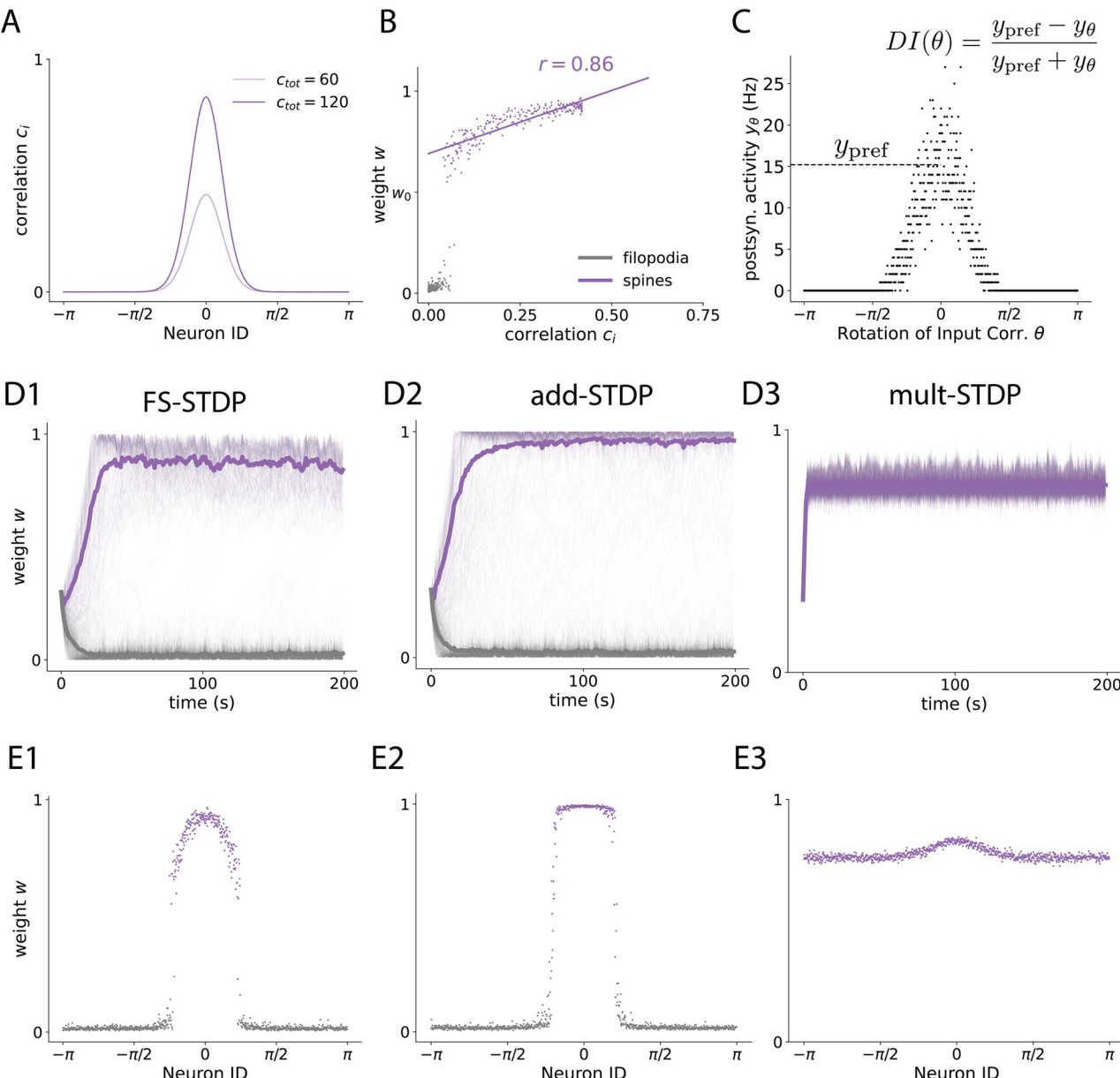

**Fig 2. Receptive Fields (RFs) formed with FS-STDP inherit the bimodality and sparsity of add-STDP, and the correlation structure representation of mlt-STDP. A**: Input correlation structure used in our stimulation protocol. Each presynaptic neuron $i$ is assigned an angle as a Neuron ID, and a correlation strength value $c_i$ according to the pdf of a von Mises distribution at that angle. $c_{tot}$ controls the total amount of correlation such that $\sum c_i = c_{tot}$ (Methods, Eq (25)). **B**: Example scatter plot of weights after learning. To obtain $r$ in Fig 3F we compute the Pearson correlation between spines' synaptic efficacy and the corresponding $c_i$ value. Grey corresponds to filopodia, and purple to spines. The straight line represents the linear regression of the synaptic efficacy after convergence of spines and the correlation strength $c_i$. **C**: Example scatter plot of neuron output activity $y_\theta$ (sampled over 1 second) for different input correlation structures. The neuron is first trained with a correlation structure centered at $\theta = \theta_{\mathrm{pref}}$. After training, the average firing rate $y_{\mathrm{pref}}$ for that correlation structure is obtained. Then, the neuron is tested for input correlations centered around the rest of angles (Rotation of Input Correlations $\theta$, also see S6 Fig), to obtain $DI(\theta)$. $DI$ (discrimination index) is obtained by averaging $DI(\theta)$ over $\theta$. To reduce the bias in $DI$, the output activity at the preferred angle ($\theta = \theta_{\mathrm{pref}}$) is measured for 100 seconds. Note how rotations in the input correlations influence the overall correlation structure but not $r_{\mathrm{pre}}$. **D**: Example of weight trajectories for FS-STDP (**D1**), add-STDP (**D2**), and mlt-STDP (**D3**). **E**: Example of receptive fields formed (weights at $t = 200$ s), same order as in **D**.

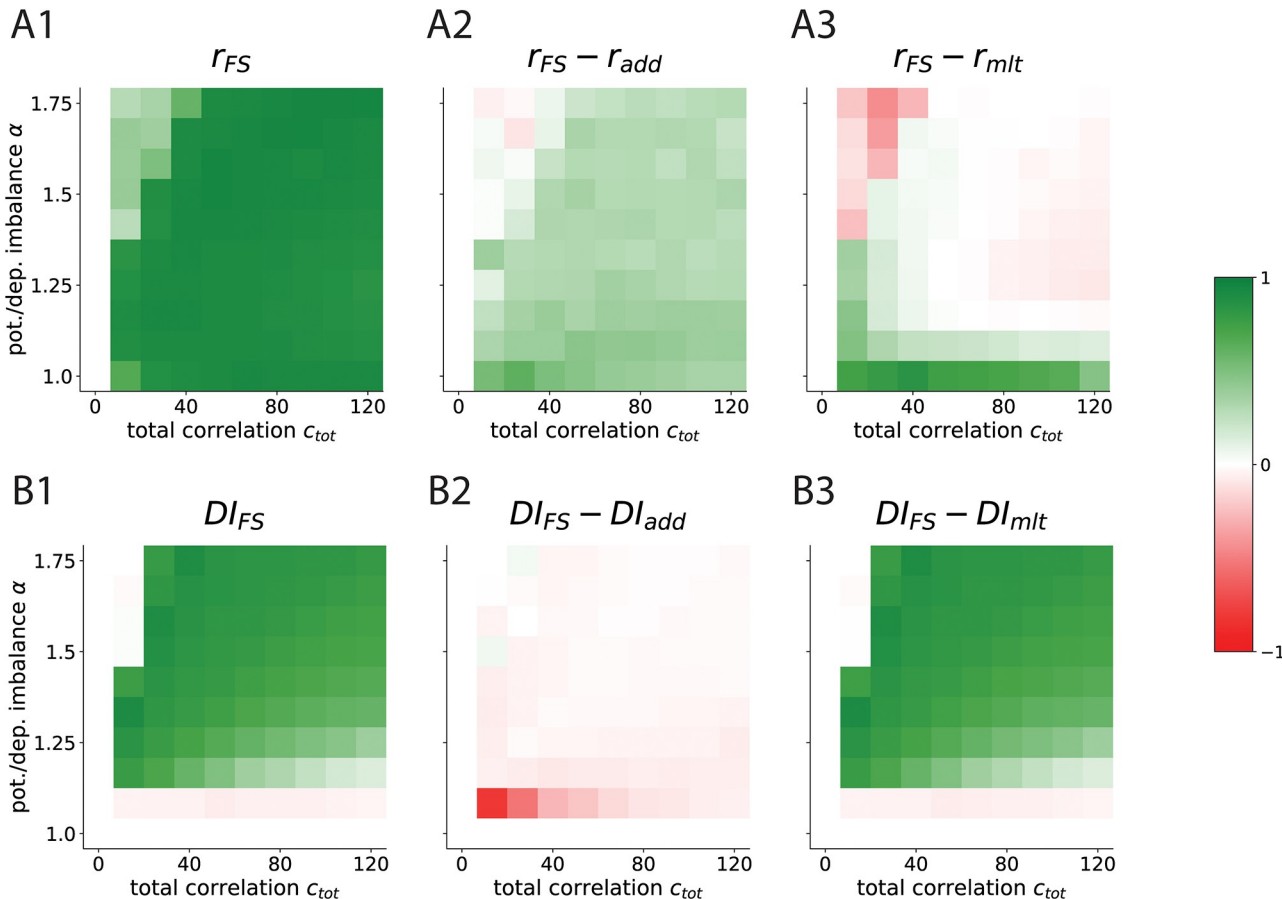

**Fig 3. Spines in STDP represent correlation structure as well as mlt-STDP, and the filopodia-spine distinction is as selective as add-STDP.**
Figure shows RF correlation representation ($r$) and discriminatory index ($DI$) across total correlation $c_{tot}$ and potentiation/depression imbalance $\alpha$ (FS-STDP and difference with add-STDP and mlt-STDP). All values are averaged over 5 seeds. **A**: Heatmaps showing the Pearson correlation for $w_i$ vs $c_i$ (see Fig 3B). Subindex labels indicate what learning rule is being shown/compared (e.g. $r_{FS} - r_{add}$ means, $r$ value for RFs obtained via FS-learning minus $r$ value obtained via additive learning). x-axis shows increasing values of total correlation and y-axis increasing values of potentiation-depression imbalance $\alpha$. Colormap ranges from Red to White to Green for values -1, 0, and 1. White regions found for small correlation values correspond to points in the parameter space where a RF has not (fully) formed. Red regions in panel F are points where a RF has formed with one rule but not the other. See S7, S8 and S9 Figs. **B**: Same for Discrimination Index averaged for all possible Neuron ID = $\theta$ (see Fig 2C).

distributions [12]. While we have compared our model to add-STDP and mlt-STDP as benchmarks of strongly competitive and weakly competitive learning, nlta-STDP itself can find a compromise between selectivity and stability with intermediate values of $\mu$. In what sense, if any, is FS-STDP different to pure nlta-STDP? nlta-STDP does offer the possibility of compromise, but precisely by finding intermediate values of either sensitivity to input correlations or specificity (which results from bimodal splits of synapses, i.e., the strong stage of the competition). In this sense, if one transitions from $\mu = 0$ (equivalent to add-STDP) to $\mu = 0.1$, one can qualitatively observe this effect (S10 Fig), where every increase in sensitivity to input correlations implies a decrease in specificity. This effect is more notable the less bimodal is the correlation structure itself. For example, while final discriminatory index in nlta and nlta* with $\mu = 0.1$ is not very different to FS-STDP with $\mu_{\text{spine}} = 0.1$ (S11 Fig) one can observe a general increase in $DI$ when there is a small background correlation (S12 Fig), where having intermediate values of $\mu$ makes the learning rule less competitive. Thus, while nlta-STDP can sometimes approximate FS-STDP, the second is more general in that it it can have an arbitrarily strong competition between filopodia and spines (governed by $\mu_{\text{filo}}$) and simultaneously make

spines compete as weakly as one desires (via $\mu_{\text{spine}}$), instead of depending on an intermediate compromise between selectivity and sensitivity.

## FS-learning makes changes resistant to new correlated input

So far, we have restricted our simulations to the case where initial weights are all the same, and salient statistical patterns in the environment have not yet been imprinted in the weight structure. Usual environments, though, don't have fixed statistics, and are subject to abrupt changes. This posits a trade-off whereby the brain needs to adapt to integrate new information, but cannot just immediately throw away any past memories. One classic solution to this problem is synaptic consolidation, which prevents memories from being erased after their formation [16–18]. In our model, synaptic consolidation is implemented with the existence of a lower soft-bound $w_0$ (Eq (4), also see Fig 1B), which makes depression approach zero when $w \rightarrow w_0$ from above. Whether this bound is effectively hard (impossible to trespass) or soft (difficult to trespass) is controlled by $\mu_{\text{spine}}$. Parameter $\mu_{\text{spine}}$ defines the approximate fixed point of $\mu$ for spines (Eqs (2) and 9). Specifically, given a consistent synaptic efficacy $w$, greater $\mu_{\text{spine}}$ values will result in higher $\mu$ values at convergence. This will, in turn, result in a *harder* soft-bound, and increase the protection of the formed receptive field. To investigate how changes in input correlations affect the preexisting synaptic structure we apply a new stimulation protocol. In this scenario, there exist two non-overlapping groups of input patterns, A and B (purple and yellow in Fig 4A, respectively). Pattern A and B are orthogonal in the sense that the neurons with nonzero correlation are disjoint. We start with pattern A for 200 seconds, and then we change the correlation structure from pattern A to pattern B for 400 seconds more. Applying this protocol, one can encounter three different scenarios: (i) the previous memory is erased with the new pattern taking over (Fig 4B2, *Total Overwriting*), (ii) the previous memory is maintained, but pattern B is also imprinted (the neuron becomes an *A or B* detector, Fig 4B3, *Partial Overwriting*), and (iii) the previous memory is maintained, and the new pattern is not imprinted (Fig 4B4, *No Overwriting*). We classify the trajectories obtained for each simulation as one of these three qualitatively different cases. To do that, we compare the RFs formed with pattern A or B alone ($\vec{w}_A$ and $\vec{w}_B$ respectively, Fig 4B1–4B4) with the one formed when first A and then B is presented $\vec{w}_{AB}$ (Fig 4C and Eq (28) in Methods). We study how the above-mentioned regimes are affected by the total amount of correlation of pattern B $c_{tot}$, the potentiation/depression imbalance $\alpha$, and $\mu_{\text{spine}}$. Higher levels of correlation in pattern B facilitate the formation of the new pattern, as the new correlated synapses can cooperate more strongly. This leads to an abundance of transitions from no overwriting to partial or total overwriting with $c_{tot}$ given a fixed $\alpha$ (Fig 4D). The effect of increasing the imbalance between potentiation and depression is not trivial, as increasing depression prevents the new memory from taking over but also increases the pressure over the previously formed memory. In consequence, and as shown in simulations, increasing $\alpha$ can drift the system from memory protection to memory overwriting but also the opposite, with the process being also coupled to the specific amount of correlation of pattern B (Fig 4D). Increasing $\mu_{\text{spine}}$ consistently protects the previously formed memory (more extended green or grey regions for higher values of $\mu_{\text{spine}}$, Fig 4D). Nevertheless, doing so also decreases the competition between filopodia and spines, thus favoring the formation of the new RF while the previous one is maintained (changes from green to grey). Compared to additive STDP, FS learning has two advantages: first, it allows a non-existing regime of memory linking/overlapping (Fig 4D4). Secondly, it allows transitioning from an add-like scenario (Fig 4D1 and 4D4) to regimes in which previously formed memories are more and more protected (Fig 4D2 and 4D3). Altogether, this shows a richness of possible interactions between previous and new environmental statistics in FS-STDP. Our learning

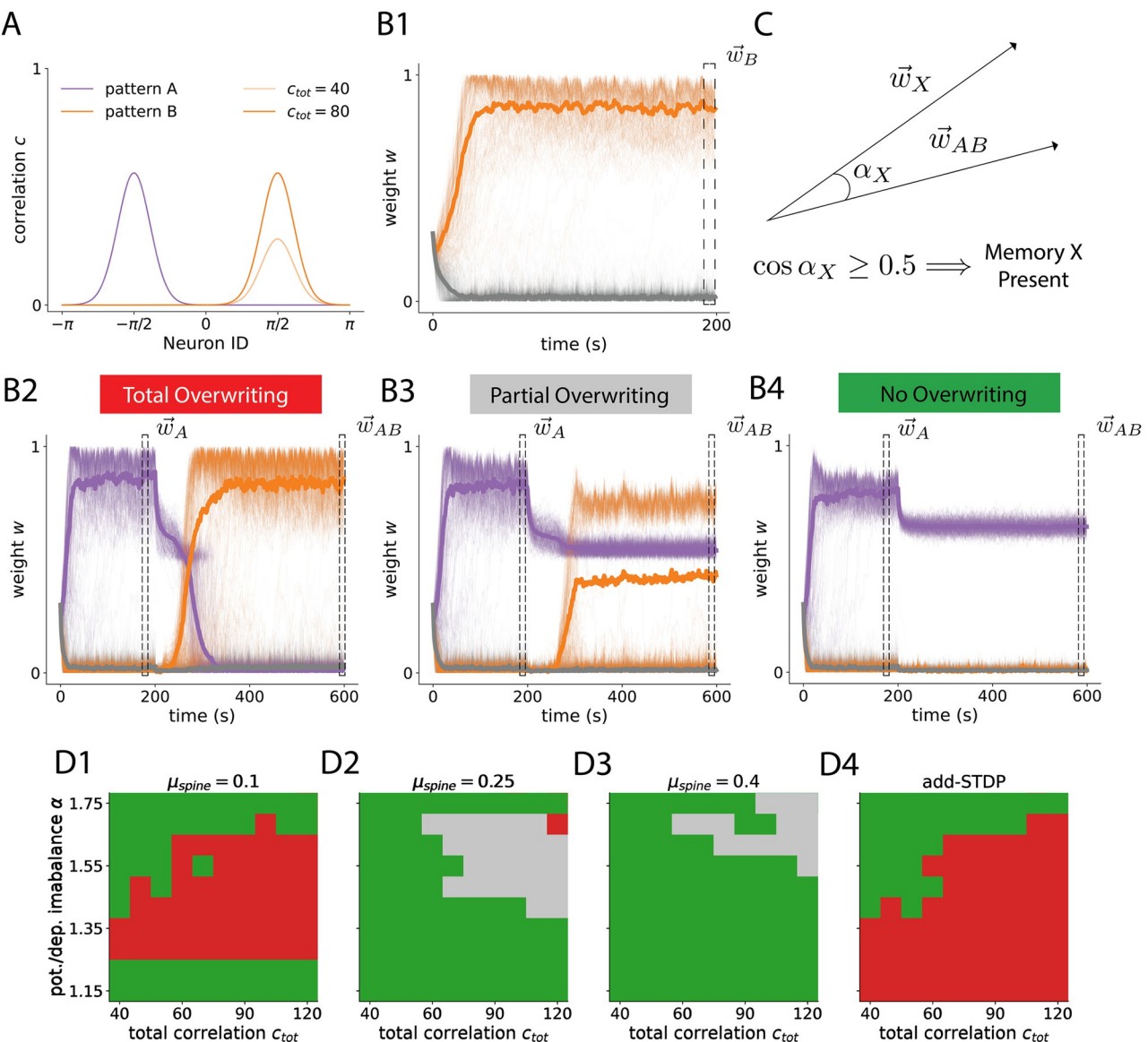

**Fig 4. FS-learning makes changes resistant to new correlated input. A**:Stimulation protocol. Input neurons have an input correlation profile as in purple (pattern A) for the first 200 seconds of the simulation. Then, it switches to the yellow correlation profile (pattern B, shifted by $\pi$ with respect to pattern A). **B**: There exist different scenarios: (i) The new memory (which would lead to **B1** in the absence of previous structure) takes over the previous one (red, **B2**). (ii) The new memory is imprinted but the old RF is conserved (grey, **B3**). (iii) The original RF is preserved and the new memory is not imprinted (green, **B4**). Orange traces correspond to weights that become spines with pattern B alone, and purple ones weights that become spines with pattern A alone. **C**: By comparing the final RF $\vec{w}_{AB}$ with that corresponding to pattern X alone (X = A,B), one can obtain a similarity score that determines whether the memory X is present or not (Methods, Eq (28)). **D**: Classification of receptive fields formed across potentiation/depression imbalance $\alpha$ and total correlation $c_{tot}$ for different protection parameter $a$ values in FS-STDP (**D1**, **D2**, and **D3**) and in add-STDP (**D4**). Colors as indicated in panels **B2**, **B3**, **B4**. Ranges of $\alpha$ and $c_{tot}$ were chosen so that a RF was always formed (see Fig 3, S7 and S8 Figs).

rule allows for an intermediate state in which receptive fields aggregate (grey), not allowed in classic models of STDP. Furthermore, it provides an active protection mechanism that goes beyond quiescence and preserves weight structure in the presence of postsynaptic activity. Interestingly, these different regimes can be controlled with model parameters like $\alpha$ or $\mu_{\text{spine}}$, so one could imagine these being physiologically driven to favor one or another depending on developmental stages or the measured environmental uncertainty.

## Discussion

We have presented a computational model that describes how filopodia and spines are differently affected by plasticity, as well as their transition from one to the other. Our learning rule, Filopodium-Spine Spike-Timing-Dependent-Plasticity (FS-STDP), posits that highly volatile and plastic filopodia exhibit strongly competitive dynamics, which we implement through an additive-like plasticity rule. In contrast, spines are assumed to follow weakly competitive dynamics, implemented via soft-bounds similar to those found in nlta-STDP with an intermediate $\mu$ or multiplicative models of STDP. We use nonlinear-temporally-asymmetric (nlta) learning to generalize the learning rules that follow both silent synapses (filopodia) and spines, with the addition of weight dependence in the parameter that governs the strength of competition of each synaptic population. FS-STDP contains two key ingredients that make it functionally appealing: (i) it is able to selectively encode, in the synaptic state, a rectified version of the correlation structure of input and (ii) it acts as a memory consolidation mechanism by protecting previously formed receptive fields. We have shown that the encoding properties of our learning rule derive from a two-stage competition (strong, followed by weak), such that synapses are first bimodally distributed (as in add-STDP) and then also continuously represent the correlation structure of the input. This is therefore an alternative solution to the *Stability vs Neuronal Specialization* dilemma [19]. Here we have focused on weights between zero and one, and have excluded the possibility of further representing the correlation structure using long-tail distributions, which could be obtained via intrinsic noise in the synaptic plasticity [20, 21]. A similar two-stage learning could be implemented by imposing a parameterized log dependence such that filopodia follow add-STDP whilst spines follow log-STDP. Our model, together with experiments, is in accord with the hypotheses that optimal network capacity can be obtained when a large fraction of the synapses are *silent* [11] and with the sparsity of network connectivity found in experiments [22]. Furthermore, FS-STDP proposes a functional explanation of how this sparse connectivity might be obtained in the first place via strongly competitive dynamics, and how it can flexibly adapt to changes in the environment statistics. In a sense, this effectively makes the first stage of FS learning a *structural plasticity rule* [4, 23]. The bimodality observed in filopodia versus spines is thus, structural, leading to an overall spine sparsity. It should be noted how this is different to a potential bimodality in the distribution of spines itself (as found in [24]).

The second property of FS-learning is the protection of formed receptive fields in the presence of new input correlations. This fits within the literature of synaptic consolidation, where it can be understood as a cascade model [16] where the *spineness* (controlled by parameter $\mu$) is an internal variable of the weight. It can also be placed in the more recent context of bidirectional dynamic processes [25], where the weight push-pulls $\mu$ and in turn high values of $\mu$ stabilize the weight. We have restricted our analysis to the interplay between fliopodia and spines, but the model could be extended to include other well-known processes of synaptic consolidation that would affect spines (as could be the experimentally [26, 27] and computationally [17, 28, 29] tested Synaptic Tagging and Capture (STC) mechanism).

FS-learning can also account for spine volatility [30]. If the $\mu$ value associated with the lower soft-bound $w_0$ leads to additive dynamics, then not all spines are protected. Instead, they decay even in the absence of a new correlated input. Decay would happen in order of previous correlation strength (less correlated decay first) until the competitive term is weak enough. This would imply an active synaptic turnover (instead of passive) making spines that are less relevant decay into filopodia first. While we do not examine directly the implementation of FS-learning in a recurrently connected network, we hypothesize that it could be beneficial for learning neuronal assemblies. While the advantages of graded synapses in memory formation via attractors have been described before [31], the representational power of FS-learning could also lead to

imprinting more complex conceptual structures. Assembly formation in recurrent networks is usually limited to independent groups of neurons that represent orthogonal inputs, as classic models of competitive learning would otherwise lead to representational collapse and the merging of assemblies. FS-STDP could potentially overcome this by distinguishing between intra-assembly and inter-assembly synaptic strengths, while maintaining a sparse connectivity. The intrinsic protection that our model gives to spines could additionally make the assemblies formed resistant to new correlated inputs that would otherwise erase them (preventing catastrophic forgetting). Our model assumes that all filopodia that reach a certain synaptic efficacy are converted into spines but, experimentally, that was found for only 9 out of 15 samples [3]. Whether the reason for this discrepancy is that (i) this transition is stochastic, (ii) the filopodia that had not become spines had not reached a high-enough synaptic efficacy, or (iii) there was an uncontrolled variable influencing this transformation remains unknown. Further studies investigating how filopodia become spines and what physiological variables affect this transformation could further constrain our model. Similarly, we assume the reverse (spines becoming filopodia) can also happen, which has not been studied yet. However, our conversion of spines into filopodia could still be considered a simplification of a renewal process in which a spine disappears and, randomly, a close-by filopodium is created. We have also focused on a simplified picture limited by the experimental results on the plasticity of filopodia, but the nature of plasticity and metaplasticity could incorporate even more states than only filopodia and spines, as can be found in the hippocampus [32]. Our study naturally suggests a systematic characterization of spike-timing-dependent plasticity in filopodia, which could help confirm if they actually follow additive dynamics and how the learning kernels vary across the filopodium-spine spectrum. This could confirm the accuracy of our implementation of filopodia dynamics and/or wether they in fact learn in a strongly competitive manner. Furthermore, we have not modelled AMPA and NMDA channels separately, while filopodia are known to contain NMDA channels even in the absence of AMPA receptors [1, 3]. We speculate that including this distinction could be beneficial for the formation of neuronal assemblies in recurrent networks, given the slow dynamics of these channels and their role as coincidence detectors [33, 34]. Finally, we leave open the question of what physiological signal(s) could represent a temporally averaged version of the synaptic strength, which in our model is called $\mu$. Since we interpret synaptic strength as the amount of AMPA receptors at the postsynaptic level, variable $\mu$ could depend on retrograde signals, which can play an important role in synaptic maturation [35]. However, it could also be dependent on other locally available signals such as synaptic volume. In summary, we have presented a model of synaptic plasticity that distinguishes between two synaptic structures: filopodia and spines. We have assumed that filopodia follow highly competitive STDP and spines learn according to soft-bounded STDP, which leads to the formation of sparse but graded receptive fields that continuously represent input correlations. Furthermore, our model proposes that the transformation of filopodia into spines can be seen as a first stage of synaptic consolidation, thus making protecting the synaptic weights learned from changes in the environmental statistics.

## Methods

### Terms and definitions

### Neuron model

We use a conductance-based Leaky Integrate-and-Fire model with excitatory and inhibitory input. Passive membrane potential dynamics are described by the following equation:

$$C_m \frac{dv(t)}{dt} = \frac{(v_{\text{rest}} - v(t))}{R_m} + g_{\text{exc}}(t)(E_{\text{exc}} - v(t)) + g_{\text{inh}}(t)(E_{\text{inh}} - v(t)) \qquad (6)$$

**Table 4. Neuron parameters.**

| Symbol | Value | Description |
|---|---|---|
| $v$ | Variable (mV) | Membrane Potential |
| $C_m$ | 200 pF | Membrane Capacitance |
| $R_m$ | 100 MΩ | Membrane Resistance |
| $v_{\text{th}}$ | -54 mV | Threshold Potential |
| $v_{\text{rest}}$ | -70 mV | Resting Potential |
| $E_{\text{exc}}$ | 0 mV | Excitatory Reversal Potential |
| $E_{\text{inh}}$ | -70 mV | Inhibitory Reversal Potential |
| $\hat{g}_{\text{exc}}$ | 0.15 nS | Excitatory Conductance Amplitude |
| $\hat{g}_{\text{inh}}$ | 0.25 nS | Inhibitory Conductance Amplitude |
| $\tau_{\text{exc}}$ | 5 ms | Excitatory Time Constant |
| $\tau_{\text{inh}}$ | 5 ms | Inhibitory Time Constant |

where the conductance response to excitatory ($g_{\text{exc}}$) and inhibitory ($g_{\text{inh}}$) presynaptic spikes is shaped using alpha functions:

$$g_{\text{A}}(t) = \hat{g}_{\text{A}} \sum_j w_j \sum_{t \geq t_j^k} (t - t_j^k) \exp[-(t - t_j^k)/\tau_A] \quad A = \text{exc, inh} \tag{7}$$

If at time $t_i^k$ neuron $i$ has membrane potential $v \geq v_{\text{th}}$, then $v$ is reset to $v_{\text{rest}}$ at $t = t_i^k + dt$, and $t_i^k$ included in the spike train $S_i(t)$. Parameter descriptions and values used in simulations are specified in Table 4.

## Synaptic plasticity model

**Filopodium-spine STDP.** Our model adds intrinsic dynamics to the parameter $\mu$ introduced in [13], making it coupled to the weight $w_i$ ($i$ denotes presynaptic index, there is only one postsynaptic neuron):

$$\frac{dw_i(t)}{dt} = \lambda \overbrace{(w_0^+ - w_i(t))^{\mu_i(t)}}^{\text{nlta}^*} \overbrace{z_i(t)S_{\text{post}}(t)}^{\text{``vanilla'' STDP}} - \lambda \overbrace{\alpha}^{\text{pot./dep. imbalance}} \overbrace{|w_i(t) - w_0^-|^{\mu_i(t)}}^{\text{nlta}^*} \overbrace{z_{\text{post}}(t)S_i(t)}^{\text{``vanilla'' STDP}} \tag{4}$$

$$\tau_\mu \frac{d\mu_i(t)}{dt} = -\left(\mu_i(t) - \frac{w_i(t) + a}{q}\right) \tag{2}$$

$S_i(t) = \sum \delta(t - t_i^k)$ are the spike trains of presynaptic neuron $i$ and $S_{\text{post}}$ the spike trains of the only postsynaptic neuron. The pre (sub-index $i$) and post (sub-index post) synaptic traces are low-pass filtered versions of the corresponding spike trains:

$$\tau_{\text{STDP}} \frac{dz_{i/\text{post}}(t)}{dt} = -z_{i/\text{post}}(t) + S_{i/\text{post}}(t) \tag{8}$$

These equations combined implement a special weight-dependence such that only small weights experience strong competition. $a$ and $q$ control what is the $\mu_i$ associated to a synaptic efficacy $w_i$ under equilibrium. In practice, we define instead $\mu_{\text{filo}}$ and $\mu_{\text{spine}}$, which indicate to what value converges a synapse with strength $w_{\text{filo}}$ and $w_{\text{spine}}$ (respectively), and then compute

the associated $a$ and $q$:

$$\begin{cases} \mu_{\mathrm{filo}} = (w_{\mathrm{filo}} + a)/q \\ \mu_{\mathrm{spine}} = (w_{\mathrm{spine}} + a)/q \end{cases} \Rightarrow \begin{cases} a = (\mu_{\mathrm{spine}} w_{\mathrm{filo}} - \mu_{\mathrm{filo}} w_{\mathrm{spine}})/(\mu_{\mathrm{filo}} - \mu_{\mathrm{spine}}) \\ q = (w_{\mathrm{filo}} + a)/\mu_{\mathrm{filo}} \end{cases} \tag{9}$$

This allows us to explicitly control the strength of the competition of filopodia and spines, by fixing what will be the typical $\mu$ values of each type of synapse. Throughout this study, we let $w_0^+ = 1$, and simply write $w_0^- = w_0$. Synaptic efficacies are clipped between $w_{\mathrm{min}} = 0$ and $w_{\mathrm{max}} = 1$. All parameters used in simulations are specified in Table 5.

**add-STDP.** We define add-STDP as in [7]:

$$\frac{dw_i}{dt} = \lambda z_i(t) S_{\mathrm{post}}(t) - \alpha \lambda z_{\mathrm{post}}(t) S_i(t) \tag{10}$$

using same parameter values as in Table 5.

**mlt-STDP.** We define mlt-STDP (mlt-STDP) as in [8]:

$$\frac{dw_i}{dt} = \lambda z_i(t) S_{\mathrm{post}}(t) - \alpha \lambda w_i z_{\mathrm{post}}(t) S_i(t) \tag{11}$$

using same parameter values as in Table 5.

**nlta-STDP.** We define nlta-STDP as in [13]:

$$\frac{dw_i(t)}{dt} = \lambda(1 - w_i(t))^\mu z_i(t) S_{\mathrm{post}}(t) - \lambda \alpha w_i(t)^\mu z_{\mathrm{post}}(t) S_i(t) \tag{12}$$

using same parameter values as in Table 5.

**Table 5. Synaptic plasticity parameters.**

| Symbol | Value | Description |
|--------|-------|-------------|
| $w_i$ | Variable (a.u.) | Synaptic Efficacy |
| $\mu_i$ | Variable (a.u.) | Spineness Parameter |
| $\mu_{\mathrm{filo}}$ | 0.01 | target $\mu$ of filopodia |
| $\mu_{\mathrm{spine}}$ | 0.1–0.4 | target $\mu$ of spines |
| $w_{\mathrm{filo}}$ | 0.1 | typical filopodium efficacy |
| $w_{\mathrm{spine}}$ | 0.75 | typical spine efficacy |
| $w_{\mathrm{min}}$ | 0 | minimum synaptic efficacy |
| $w_{\mathrm{max}}$ | 1 | maximum synaptic efficacy |
| $\lambda$ | 0.006 a.u. | Learning Rate |
| $\alpha$ | 1.00–1.75 a.u. | Potentiation/Depression Imbalance |
| $w_0^+$ | 1 a.u. | Higher Soft-Bound |
| $w_0^-$ | 0.5 a.u. | Lower Soft-Bound |
| $S_i$ | Variable (Hz) | Presynaptic Spike Train (neuron index = $i$) |
| $S_{\mathrm{post}}$ | Variable (Hz) | Postsynaptic Spike Train |
| $\tau_{\mathrm{STDP}}$ | 20 ms | STDP time constant |
| $z_i$ | Variable (a.u.) | Presynaptic Trace |
| $z_{\mathrm{post}}$ | Variable (a.u.) | Postsynaptic Trace |
| $\tau_\mu$ | 20 s | $\mu$ time constant |

**nlta\*-STDP.** We define nlta\*-STDP as FS-STDP without $\mu$ dynamics, or equivalently, as nlta-STDP incorporating arbitrary soft-bounds $w_0^+$ and $w_0^-$

$$\frac{dw_i(t)}{dt} = \lambda(w_0^+ - w_i(t))^\mu z_i(t)S_{\text{post}}(t) - \lambda\alpha|w_i(t) - w_0^-|^\mu z_{\text{post}}(t)S_i(t) \tag{13}$$

using same parameter values as in Table 5.

## Mean synaptic dynamics in a linear Poisson neuron

Following [13], we recover some of the main results assuming a Poisson linear neuron, which allows for an exact solution of the weight mean-field dynamics. In this context, the output neuron is the realization of an inhomogeneous Poisson process with an instantaneous firing rate

$$R^{\text{post}}(t) = \frac{1}{N_{\text{pre}}}\sum_i w_i(t)S_i(t - \epsilon) \tag{14}$$

where $N_{\text{pre}}$ is the number of presynaptic neurons and $\epsilon$ a small constant delay. Under these conditions, and if one has a stimulation protocol without backward correlations (as is our case, see below), one can obtain the mean-field description of the time evolution of $w_i$:

$$\dot{w}_i = \frac{\lambda\tau r_{\text{pre}}^2}{N_{\text{pre}}}\left[-\Delta f(w_i)\sum_j w_j + f_+(w_i)\sum_j w_j C_{ij}^+\right] \tag{15}$$

where $r_{\text{pre}}$ is the presynaptic firing rate, $f_+(w_i)$ and $f_-(w_i)$ are the time-difference-independent amplitudes of potentiation and depression (respectively) and $\Delta f(w_i) \equiv f_-(w_i) - f_+(w_i)$. $C_{ij}^+$ is called the *integrated normalized cross-correlation* and is a function of the *normalized cross-correlation* $\Gamma_{ij}^0(\Delta t)$, and are defined as:

$$\Gamma_{ij}^0(\Delta t) \equiv \frac{\langle S_i(t)S_j(t + \Delta t)\rangle_t}{r_{\text{pre}}^2} - 1 \tag{16}$$

$$C_{ij}^+ \equiv \int_0^\infty d\Delta t \frac{1}{\tau}K(\Delta t)\Gamma_{ij}^0(\Delta t - \epsilon) \tag{17}$$

with $K(\Delta t)$ the *learning kernel*, which defines how the weight changes as a function of the spike-time difference between pre and postsynaptic spikes, and is usually an exponential decay of the absolute difference of spike times.

## Competition and cooperation factors in models of STDP

For completeness, we include in Table 6 the competition and cooperation factors (see previous section and also Table 3) associated to each model of STDP mentioned in this study:

## Generating temporally correlated spike trains

To correlate a pool of presynaptic Poisson neurons, we generate a reference spike train:

$$\rho_{\text{ref}}(t) = \sum \delta(t - t_{\text{ref}}^k) \tag{18}$$

via a homogeneous Poisson process: $P(t \in \{t_{\text{ref}}^k\}) \equiv P(X_t^{\text{ref}} = 1) = r_{\text{pre}}\Delta t$ ($\Delta t$ a small timestep). Now, for each neuron $i$ in with correlation strength $c_i$, the probability of firing at time $t$

**Table 6. Competition and cooperation factors across models of STDP.**

| Rule | cooperation factor $f_+(w)$ | competition factor $\Delta f_-(w)$ |
|---|---|---|
| add-STDP | 1 | $1 - \alpha$ |
| mlt-STDP | 1 | $1 - w\alpha$ |
| mlt/mlt-STDP | $1 - w$ | $1 - w(1 + \alpha)$ |
| nlta-STDP | $(1 - w)^\mu$ | $(1 - w)^\mu - \alpha w^\mu$ |
| nlta*-STDP | $(1 - w)^\mu$ | $(1 - w)^\mu - \alpha|w - w_0|^\mu$ |
| FS-STDP | $(1 - w)^{\mu(t)}$ | $(1 - w)^{\mu(t)} - \alpha|w - w_0|^{\mu(t)}$ |

depends on whether that time is included in the reference spike train or not:

$$P(X_t^i = 1|X_t^{\text{ref}} = 1) = r\Delta t + \sqrt{c_i}(1 - r\Delta t) \tag{19}$$

$$P(X_t^i = 1|X_t^{\text{ref}} = 0) = r\Delta t(1 - \sqrt{c_i}) \tag{20}$$

This ensures an average firing rate $r$ over the defined period of time, while also imposing the following instantaneous cross-correlations:

$$\Gamma_{ij}^0(\Delta t) = \frac{1}{r}c_{ij}\delta(\Delta t) \tag{21}$$

where $c_{ij}$ denotes the pair-wise correlations

$$c_{ij} = \frac{\text{Cov}(X_t^i, X_t^j)}{\sqrt{\text{Var}(X_t^i)\text{Var}(X_t^j)}} = \sqrt{c_i c_j} \tag{22}$$

In turn, this results in the following integrated normalized cross-correlations, which define the level of cooperation between synapses:

$$C_{ij}^+ = \frac{\sqrt{c_i c_j}}{\tau r} \tag{23}$$

Note how, opposed to [13], here we have generalized to an arbitrary correlation structure $\vec{c}$, so not all neurons within a correlated pool are necessarily equally correlated.

## von Mises shaped correlation

The von Mises distribution is often used to describe tuning to different inputs that have a circular or periodic relationship. Its density function over an angle $\theta$ is given by

$$f_{\text{vM}}(\theta|\theta_{\text{pref}}, \kappa) = \frac{\exp\left(\kappa \cos(\theta - \theta_{\text{pref}})\right)}{2\pi I_0(\kappa)} \tag{24}$$

with $I_0(\kappa) = \int \exp(\kappa \cos \theta)d\theta$. $\theta_{\text{pref}}$ denotes the center of the distribution (or preferred angle) and $\kappa$ controls its width (or variance). Given a Neuron ID $= \theta_i$, we assign each presynaptic neuron a correlation strength $c_i = f_{\text{vM}}(\theta_i)$ and then normalize according to parameter $c_{\text{tot}}$ such that:

$$c_{\text{tot}} = \sum c_i \tag{25}$$

Parameter $c_{\text{tot}}$ gives a measure of the total drive that the postsynaptic neuron effectively receives due to presynaptic temporal correlations.

## Squared pulse correlations

Squared pulse correlation structures are 0 for every neuron except for a number $N_{tot} = 200$ neurons. Correlated neurons are assigned a correlation value $c_i = c_{tot}/N_{tot}$.

## Discrimination index

We define $y_\theta$ as the firing rate of the output neuron in the presence of an input correlation structure centered at $\theta$. Then, the *Discrimination Index* at angle $\theta$ ($DI(\theta)$), of a neuron trained with input centered at $\theta = $ pref, is

$$DI(\theta) = \frac{y_{\text{pref}} - y_\theta}{y_{\text{pref}} + y_\theta} \tag{26}$$

We then define the general Discrimination Index ($DI$) as:

$$DI = \frac{1}{|\text{Neuron ID}|} \sum_{\theta \in \text{Neuron ID}} DI(\theta) \tag{27}$$

That is, the discrimination index evaluated at $\theta$ averaged over all possible values of $\theta$.

## Memory overlap

Given a set of conditions $X$, and a set of conditions $Y$, the *memory* after convergence is defined as $\vec{w}_X$ and $\vec{w}_Y$ (respectively). Then, we measure the overlap between both memories via their cosine similarity:

$$\cos(\alpha_{XY}) = \frac{\langle \vec{w}_X, \vec{w}_Y \rangle}{||\vec{w}_X|| \cdot ||\vec{w}_Y||} \tag{28}$$

where $\langle \vec{a}, \vec{b} \rangle$ is the dot product of vectors $\vec{a}$ and $\vec{b}$, $||\vec{a}||$ is the norm $\sqrt{\langle \vec{a}, \vec{a} \rangle}$ of $\vec{a}$, and $\alpha_{XY}$ is the angle between $\vec{w}_X$ and $\vec{w}_Y$. We use a threshold of 0.5 to determine whether a memory is present or not. The threshold was chosen such that the colormap in Fig 4D qualitatively corresponds with the regimes presented in Fig 4B2, 4B3 and 4B4 (total, partial or no overwriting). Also see S13, S14, S15 and S16 Figs.

## Implementation details

We use BRIAN2 [36] in our simulations. Additional implementation parameters used in simulations (not specified in the Figures or Figure Captions) can be found in Table 7.

**Presynaptic firing rates.** Inhibitory neurons have a fixed 10 Hz firing rate. For excitatory neurons, we use a presynaptic firing rate ($r_{\text{pre}}$) of 30 Hz in all our simulations except for computing the Discrimination Index ($DI$). A relatively high (30 Hz) presynaptic rate speeds-up simulations and avoids quiescence modes with an absence of plasticity. For example, in the simulations of Fig 4, a low presynaptic rate would lead to an effective *No Overwriting* regime because the postsynaptic neuron is silent and the weights are frozen. However, we are interested in the case where the learning rule inherently protects the RF even with ongoing neuronal activity and synaptic plasticity. At high presynaptic firing rates, however, the postsynaptic neuron firing rate does not depend on the selectivity of the RF to the presented pattern. This is because the neurons with non-zero weight are able to excite the postsynaptic neuron even in the absence of correlations. To use the $DI$ as a selectivity measure, which has the advantage of being normalized, we use a presynaptic firing rate of 10 Hz once the RF has formed.

**Use of total correlation $c_{\text{tot}}$ and potentiation/depression imbalance $\alpha$ as parameter space.** If one takes the mean-field learning dynamics (Eq (15)) and substitutes the cross-

**Table 7. Simulation parameters.**

| Symbol | Value | Description |
|---|---|---|
| $\Delta t$ (int.) | 0.5 ms | Integration timestep |
| $\Delta t$ (input) | 1 second | Input timed array width |
| $\Delta t$ ($w$ recording) | 1 second | Weight recording timestep |
| $w$ (initial) | 0.3 a.u. | Initial synaptic efficacy |
| $\kappa$ | 8 a.u. | von Mises pdf width parameter |
| $r_{\mathrm{pre}}$ (default) | 30 Hz | Presynaptic firing rate (training) |
| $r_{\mathrm{pre}}$ (discrimination index) | 10 Hz | Presynaptic firing rate (testing) |
| $r_{\mathrm{pre}}$ (inhibition) | 10 Hz | Presynaptic firing rate (inhibitory neurons) |

| Figure | Variable | Value |
|---|---|---|
| Fig 1 | $\alpha$ | 1.35 |
| Fig 1 | $\mu_{\mathrm{spine}}$ | 0.1 |
| Fig 1 | $c_{\mathrm{tot}}$ | 60 |
| Fig 2 | $\mu_{\mathrm{spine}}$ | 0.1 |
| Fig 2 | $\alpha$ | 1.35 |
| Fig 2B, 2C, 2DX and 2EX | $c_{\mathrm{tot}}$ | 60 |
| Fig 3AX and 3BX | $\mu_{\mathrm{spine}}$ | 0.1 |
| Fig 4BX | $\alpha$ | 1.35 |
| Fig 4BX | $c_{\mathrm{tot}}$ (pattern A and pattern B) | 60 |
| Fig 4B1 and 4B2 | $\mu_{\mathrm{spine}}$ | 0.1 |
| Fig 4B3 | $\mu_{\mathrm{spine}}$ | 0.15 |
| Fig 4B4 | $\mu_{\mathrm{spine}}$ | 0.3 |

| Supporting Information Figures | | |
|---|---|---|
| S1, S2 and S3 Figs | Same as 1G | |
| S4, S5, S7, S8, S9 and S11 Figs | Same as Fig 3 | |
| S10 | Same as Fig 2 | |
| S12 | Same as Fig 3 (with background correlations) | |
| S13, S14, S15 and S16 Figs | Same as Fig 4 | |

correlations $C_{ij}^{+}$ obtained in Eq (23), the following expression is obtained:

$$\dot{w}_i = \frac{\lambda \tau r_{\mathrm{pre}}^2}{N} \left[ \overbrace{-\Delta f(w_i)}^{\text{linear in } \alpha} \sum_j w_j + \overbrace{\frac{c_{\mathrm{tot}} f_+(w_i)}{r_{\mathrm{pre}}}}^{\text{linear in } c_{\mathrm{tot}}} \sum_j w_j \frac{\sqrt{c_i^{\mathrm{norm}} c_j^{\mathrm{norm}}}}{\tau} \right] \tag{29}$$

where $c_i^{\mathrm{norm}}$ are the normalized correlation values ($\sum_i c_i^{\mathrm{norm}} = 1$), which are fixed given $\theta_{\mathrm{pref}}$ and $\kappa$. One can see that the effect of increasing $r_{\mathrm{pre}}$ is equivalent to: (i) quadratically increasing the learning rate and (ii) inversely decreasing cooperation. Instead of changing $r_{\mathrm{pre}}$ (which would have with the aforementioned inconveniences), we sweep over the parameters $\alpha$ and $c_{tot}$. This allows exploring different ratios of competition (modulated by $\alpha$) and cooperation (modulated by $c_{\mathrm{tot}}$), while yielding similar stability and training times.

## Supporting information

**S1 Fig. Example trajectories of synaptic efficacies (top), mean-field interactions (middle) and competition/cooperation factors (bottom), using a correlation structure sampled from a Gaussian (same as Fig 1).** Obtained for add-STDP, nlta\*-STDP and FS-STDP (left to right). As usual, gray means filopodia in the case of FS-STDP, and simply $w_i > 0$ for the rest of learning rules. Note how grey curves of FS-STDP approximate those of add-STDP, and purple ones those of nlta\*-STDP. $\mu = 0.91$ (for nlta\*-STDP) taken as the average $\mu$ of spines obtained via FS-STDP.
(TIF)

**S2 Fig. Same as S1 Fig, but with a squared pulse correlation structure (all synapses equally correlated and the rest with 0 correlation).** $\mu = 0.12$ (for nlta\*-STDP) taken as the average $\mu$ of spines obtained via FS-STDP.
(TIF)

**S3 Fig. Same as S1 Fig, but with a von Mises correlation structure.** This structure is used throughout the paper for its similarity to a squared pulse (used in most previous studies), its yet rich structure (not exaclty binary) and the plausibility of these type of correlations in visual cortex. $\mu = 0.11$ (for nlta\*-STDP) taken as the average $\mu$ of spines obtained via FS-STDP.
(TIF)

**S4 Fig. Receptive Fields after convergence for a squared pulse correlation structure (FS-STDP).** $c_{\text{tot}}$ and $\alpha$ as in the 10x10 pixels of heatmaps in Fig 3.
(TIF)

**S5 Fig. Receptive Fields after convergence for a squared pulse correlation structure (add-STDP).** $c_{\text{tot}}$ and $\alpha$ as in the 10x10 pixels of heatmaps in Fig 3.
(TIF)

**S6 Fig. Heatmap showing the correlation of every neuron (by Neuron ID) with rotation of input correlations $\theta$.**
(TIF)

**S7 Fig. Receptive Fields after convergence for FS-STDP.** Each subpanel corresponds to each of the 10x10 pixels in the heatmaps shown in Fig 3A1 and 3B1.
(TIF)

**S8 Fig. Receptive Fields after convergence for add-STDP.** Each subpanel corresponds to each of the 10x10 pixels in the heatmaps shown in Fig 3A2 and 3B2.
(TIF)

**S9 Fig. Receptive Fields after convergence for mlt-STDP.** Each subpanel corresponds to each of the 10x10 pixels in the heatmaps shown in Fig 3A3 and 3B3.
(TIF)

**S10 Fig. Example of nlta-STDP receptive fields from $\mu = 0$ to $\mu = 0.1$.**
(TIF)

**S11 Fig. Same as Fig 3, but comparing FS-STDP (Left) to nlta-STDP both in its standard form (Center) and with a lower bound $w_0$ as in FS-STDP (nlta\*-STDP, Right).** $\mu = 0.075$.
(TIF)

**S12 Fig. Same as S11 Fig, but with a correlation increase of 0.05 in every presynaptic neuron with respect to S11 Fig (then renormalized).**
(TIF)

**S13 Fig. Receptive Fields after convergence for FS-STDP, $\mu_{spine} = 0.1$, (pattern A and then pattern B).** Each subpanel corresponds to each of the 10x10 pixels in the heatmaps shown in Fig 4C1.
(TIF)

**S14 Fig. Receptive Fields after convergence for FS-STDP, $\mu_{spine} = 0.25$, (pattern A and then pattern B).** Each subpanel corresponds to each of the 10x10 pixels in the heatmaps shown in Fig 4C2.
(TIF)

**S15 Fig. Receptive Fields after convergence for FS-STDP, $\mu_{spine} = 0.4$, (pattern A and then pattern B).** Each subpanel corresponds to each of the 10x10 pixels in the heatmaps shown in Fig 4C3.
(TIF)

**S16 Fig. Receptive Fields after convergence for add-STDP (pattern A and then pattern B).** Each subpanel corresponds to each of the 10x10 pixels in the heatmaps shown in Fig 4C4.
(TIF)

## Author Contributions

**Conceptualization:** Albert Albesa-González, Claudia Clopath.

**Formal analysis:** Albert Albesa-González.

**Funding acquisition:** Claudia Clopath.

**Investigation:** Albert Albesa-González, Claudia Clopath.

**Methodology:** Albert Albesa-González, Claudia Clopath.

**Software:** Albert Albesa-González.

**Supervision:** Claudia Clopath.

**Writing – original draft:** Albert Albesa-González.

**Writing – review & editing:** Claudia Clopath.

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
