## [Decision Letter · Decision Letter 0]

31 Oct 2023

Dear Mr. Albesa Gonzalez,

Thank you very much for submitting your manuscript "Learning with filopodia and spines: complementary additive and multiplicative STDP lead to specialized, graded and protected receptive fields." for consideration at PLOS Computational Biology.

As with all papers reviewed by the journal, your manuscript was reviewed by members of the editorial board and by several independent reviewers. In light of the reviews (below this email), we would like to invite the resubmission of a significantly-revised version that takes into account the reviewers' comments.

The reviewers have raised some very substantial concerns, particularly reviewer 3. Most important within those concerns raised are (1) the lack of accuracy in the application of the term "multiplicative weight update", (2) the misrepresentation of past work in this area, and (3) the lack of clarity that the model is not a mechanistic model of filopodia and spines, and is really more of a phenomenological hypothesis about their roles in learning.

We cannot make any decision about publication until we have seen the revised manuscript and your response to the reviewers' comments. Your revised manuscript is also likely to be sent to reviewers for further evaluation.

Sincerely,

Blake A Richards

Academic Editor

PLOS Computational Biology

Daniele Marinazzo

Section Editor

PLOS Computational Biology

The reviewers have raised some very substantial concerns, particularly reviewer 3. Most important within those concerns raised are (1) the lack of accuracy in the application of the term "multiplicative weight update", (2) the misrepresentation of past works in this area, and (3) the lack of clarity that the model is not a mechanistic model of filopodia and spines, and is really more of a phenomenological hypothesis about their roles in learning.

Reviewer's Responses to Questions

**Comments to the Authors:**

Reviewer #1: General comment: This paper is very interesting. It extends Gutig et al, 2003 with a time-varying, weight-dependent mu, and it relates the model to recent experimental findings suggesting that weak filopodia synapses are subject to a different learning rule than stronger spine synapses. I really like the concept, and I found the empirical results intuitive given the explanations and visualizations provided. However, the presentation (explanations in text and visualizations in figures) of how the learning rule works and how it produces the empirical results could be improved in order to be digested by a larger audience of theorists and experimentalists.

Suggestions for improvement:

P1L36: sp - *depolarized

P2L54-59: What experimental results show unimodal weight distributions? Isn’t selectivity typically associated with skewed distributions?

Buzsaki, G., Mizuseki, K., 2014. The log-dynamic brain: how skewed distributions affect network operations. Nat Rev Neurosci 15, 264–78. doi:10.1038/nrn3687

This study found bimodal distributions:

Dorkenwald, S., Turner, N.L., Macrina, T., Lee, K., Lu, R., Wu, J., Bodor, A.L., Bleckert, A.A., Brittain, D., Kemnitz, N., Silversmith, W.M., Ih, D., Zung, J., Zlateski, A., Tartavull, I., Yu, S.-C., Popovych, S., Wong, W., Castro, M., Jordan, C.S., Wilson, A.M., Froudarakis, E., Buchanan, J., Takeno, M.M., Torres, R., Mahalingam, G., Collman, F., Schneider-Mizell, C.M., Bumbarger, D.J., Li, Y., Becker, L., Suckow, S., Reimer, J., Tolias, A.S., Costa, N.M. da, Reid, R.C., Seung, H.S., 2022. Binary and analog variation of synapses between cortical pyramidal neurons. eLife 11, e76120. doi:10.7554/elife.76120

P3L98-99: Might be nice to have a Supplementary Figure where you show some example traces of dynamics for a few synaptic weights and their corresponding mu parameter during learning.

P3L109: At this point in the text, can you reference where in the equations provided in the methods this competition is implemented? Eq. 3 and 4 describe how one weight evolves independently. Both the potentiation or depression components depend on pre-post correlations, so intuitively, if two presynaptic inputs were correlated with each other, they would share a similar correlation with the output/trace of the postsynaptic neuron, and undergo a similar weight update. However, changing one weight changes the firing rate of the postsynaptic neuron, which then influences other inputs with varying degrees of correlation with each other and with the postsynaptic neuron. This intuition should be provided in the text. Eq. 9 that shows that there is a depression component that is independent of cross-correlations. To understand Eq. 9, one needs to find Eq. 12 and 13. The manuscript as is makes it very difficult to find all the information one needs to understand Fig. 1.

P5, Fig. 1C-E: I struggled here. To understand how the learning rule in Eq. 3 and 4 relates to the terms “competition,” “cooperation,” “push,” and “pull,” I really had to go back and forth between the Eq. 3, 4, 9, 12 and 13, reading the text, and staring at Fig. 1C-E. Something needs to be improved for the naïve reader here.

P5, Fig. 1E: What values of the ‘a’ and ‘alpha’ parameters do you assume in this panel? Only a large range is given in Table 2.

P5, Fig. 1C-E: There are really two meanings of the word “competition” used in this paper.

The first one is the most important one and relates to the empirical results in Fig. 2E. The end product of an additive STDP rule with weight bound is a binary distribution of weights that do not capture the structure of presynaptic correlations. This is referred to as “strongly competitive.” The intuition for this is that whichever inputs strengthen early, increase the firing rate of the postsynaptic cell, which then increases their correlations with the postsynaptic cell, and they further potentiate. Meanwhile, inputs that are not well correlated with the postsynaptic cell experience a post-synaptic trace-dependent depression (Eq. 3). On the other hand, the end product of a multiplicative STDP rule with weight bound is a unimodal distribution of weights, where the weight co-varies with the degree of correlation with the postsynaptic cell. This is referred to as “weakly competitive” – everyone’s allowed a relatively strong weight. However, the baseline weight is elevated, and the signal-to-noise ratio is poor, leading to poor discriminability between stimuli. The hybrid model gets the best of both worlds. It is this meaning of the term “competitive” that is used in Fig. 1C (filopodia are strongly competitive, and spines are weakly competitive). This could be better explained in the text (how to get from Eq. 3 to Eq. 9, 12, 13).

The second use of the term “competitive” refers to a specific term within Eq. 9 and 12 that falls out of the learning rule (Eq. 3) when firing rate is computed as a weighted sum of inputs. In Figure 1D, this “competitive” force is presented as being “depressive.” This caused me as a reader to look at the depression term in Eq. 3, which led to confusion. What’s worse is that this “competitive” term in Eq. 9 and 12 can actually switch signs! When it is positive, it is depressing, but when it is negative, it is actually potentiating! I know that this use of the term “competitive” and also the use of the term “cooperative” came directly from Gutig et al., but that doesn’t they are best for building intuition about this model! Fig. 1C and 1E present conflicting intuitions due to the dual meaning of the term “competitive”. In C, it is stated that spines (strong weights) “compete” weakly with each other. However, in E, it is shown that the “competitive” term (blue) is stronger for spines! It is only weak for spines close to w0 (in fact it goes negative, turning into “cooperativity”/potentiation). The “cooperative” term (red) is shown as decreasing for weights between w0 to 1. Both of these, at face value, imply that spines are more “competitive”/depressing than filopodia, not less.

The inverse: in C, it is stated that filopodia (weak weights) “compete” strongly with each other. However, in 1E, weights below w0 have strong “cooperation” (red) and low “competition” (blue), so their net effect should be cooperative/potentiating?

I have an even tougher time understanding “push” and “pull,” because panels D and E don’t show how weak weights influence strong weights or visa versa. Does a weak filopodia “pull” a strong spine towards it, which would be down or depressive? Does a strong spine “push” a weak filopodia away from it, which would also be down/depressive? Please enlighten the reader.

P5, Fig. 1E: Here’s a suggestion. What if instead of labeling the two terms in Eq. 9 as “competition” and “cooperation,” what if they were instead labeled “correlation-independent interaction” and “correlation-dependent interaction” ? With a sign flip, the correlation-independent interaction could be depressive when that term is negative, and it could be potentiating when that term is positive. Whether or not you adapt this suggestion, I hope you can see that something needs to be made less confusing in Fig. 1 to really grok the key concepts.

P3L125: Reference Eq. 9.

P4L160: I think what you’ve done here is created presynaptic inputs that have visual receptive fields in orientation space. There is a “reference” input that has a von Mises/Gaussian shaped receptive field over orientations. Each input has identical constant firing rates. If each presynaptic neuron is assigned a different preferred orientation, then it will also have a different cross-correlation with the reference neuron. Then in Fig. 2C, I think what you are doing is taking one stimulus orientation, and “showing” it to the network, so that you get a distribution of firing rates over the inputs, and this gets filtered through the learned weights, producing a post-synaptic firing rate for each orientation. It’s confusing that the x-axis is labeled “Neuron ID” rather than “Stimulus orientation.” If this is the case, can you please describe it that way? If not, can you clarify? There is no supplementary section explaining this further. Can you show population heat maps of the presynaptic firing rates versus orientation for two different values of c_total?

P4L183: “formation *of the RFs”

P7, Fig. 2B: “Figure F,” there is no panel F.

P4L165 and P7, Fig. 2B: “discrimination index DI” – should this be reported in panel C? Panel C needs clarification in the legend. Y_pref is not described.

P4L203: sp - *throw

P6L215: “*at then”

P6L233: “consistently *protects” ?

P6L234: There are no blue regions in Fig. 4D. Do you mean grey?

P6L236: There are no orange regions in Fig. 4D. Do you mean “changes from red to grey” ?

P6L243: *orange again

Reviewer #2: Based on recent experimental observations, the authors developed a computational model implementing a novel synaptic plasticity rule based on STDP-like mechanisms for changes in postsynaptic filopodia (as a structural correlate of silent synapses) and spines (as a structural correlate of non-silent synapses). A major component of this plasticity and learning rule is a strong competition of filopodia to be converted into spines (related to additive plasticity), and a weak competition of spines to encode the representation of input correlations (related to multiplicative plasticity). The filopodium-spine learning rule is elegant because it is formalized using nonlinear-temporally-asymetric learning with just one parameter μ affecting its additivity and multiplicativity (with filopodia/spines having small/large μ values, respectively). Interestingly, the learning rule prevents the disruption of previously learned receptive fields after the emergence of new input correlations - thus supporting memory consolidation. Overall, this computational work represents a solid and innovative contribution to the field.

Comments:

Figure 1G2 and Lines 264-266: 'have excluded the possibility...' and 'imposing a parameterized log dependence such that filopodia follow add-STDP and spines log-STDP'

It is not obvious that a skewed distribution is necessary to represent correlations; it tends to be a conserved feature of synaptic weights and spine sizes (cf. Turrigiano et al., Nature 1998; Hazan & Ziv, J Neurosci 2020).

This point – especially regarding the activity-independence of the skewed weight distribution - could be expanded, particularly given that the distribution of spine weights in Figure 1G2 (but not filopodia weights) does not have the experimentally observed skew with long tails. Although the authors mention activity-dependent plasticity rules (e.g. log-STDP) as a potentially relevant mechanism, the lognormal-like distribution of spine sizes (and perhaps also filopodia sizes) emerges even in the complete absence of synaptic transmission– as a result of so-called “intrinsic” (i.e. activity-independent) synaptic dynamics (Hazan & Ziv 2020, Rößler et al. Open Biol 2023). Therefore, recent computational models of lognormal-like (skewed) synaptic weight changes have been based either on purely intrinsic fluctuations of synapse/spine sizes (Hazan & Ziv 2020, Eggl et al. Comm Biol 2023, Rößler et al. 2023) or on a combination of extrinsic (activity-dependent) and intrinsic dynamics (Rößler et al. 2023) represented as STDP combined with (multiplicative) noise, respectively.

How would the inclusion of such “intrinsic” fluctuations in spine (and perhaps filopodia) size affect the main results of the manuscript? Could the authors explore this by including, for example, multiplicative noise as a mechanism for intrinsic synaptic dynamics? If this requires an extensive re-tuning and re-simulation of the model then the authors could at least discuss this in the text (perhaps as an outlook).

Minor:

---

## [Decision Letter · Decision Letter 1]

9 Apr 2024

Dear Mr. Albesa Gonzalez,

Thank you very much for submitting your manuscript "Learning with filopodia and spines: complementary strong and weak competition lead to specialized, graded, and protected receptive fields." for consideration at PLOS Computational Biology. As with all papers reviewed by the journal, your manuscript was reviewed by members of the editorial board and by several independent reviewers. The reviewers appreciated the attention to an important topic. Based on the reviews, we are likely to accept this manuscript for publication, providing that you modify the manuscript according to the remaining minor review recommendations.

Sincerely,

Blake A Richards

Academic Editor

PLOS Computational Biology

Daniele Marinazzo

Section Editor

PLOS Computational Biology

Reviewer's Responses to Questions

**Comments to the Authors:**

Reviewer #1: The authors have addressed my previous concerns. Below are some additional minor issues that should be corrected, but the manuscript is otherwise ready for publication.

P3L131 – typo: “in the of weight dependence”

P5L148 – not sure if Eq. (2) needs to be repeated as Eq. (5)

P7, Fig. 1E legend – are the grey shaded areas for u in [0, 0.1], and the purple areas for u in [0.5, 1] ? Please indicate this in the legend.

P8L16 – do you mean mu_spine here?

P9, Fig. 2 – In the previous review, I asked for clarification about the stimulus. I understand it better now (tell me if I’m wrong). All units are firing at the same time-averaged rate. Some units are highly correlated with a reference spike train (von Mises theta = 0), and others have correlations that drop off with increasing von Mises angle theta. So it is definitely not like an oriented bar visual stimulus being presented, where different neurons would have different degrees of correlation because they have different “orientation preferences”. I know I requested the xlabel “Stimulus Orientation” for Fig. 2C, but now that I understand that it is the correlations of the inputs that are being rotated, perhaps a more precise xlabel would be “Rotation of Input Correlations (theta)”.

P11L382 – typo: “biomdally”

P12, Fig. 4D – For the grey regions where memory overlap is allowed, wouldn’t the discriminability of A vs. B be poor? I guess the postsynaptic cell would now be an “A or B” detector, and still discriminate vs. other correlation rotations? Ideally this would be discussed in the paper.

P13L401 – grammar: “high values [of] mu”

P14L466 – typo: “small(s)”

P14L468 – one of these should be mu_filo

P15, Table 2: mu_spine is labeled mu_filo

P17L521 – typo: “(or) preferred angle”

Reviewer #2: I think the authors have addressed the main points of the reviewers.

Authors should make the computational code underlying their results fully available. I did not notice this in the revised manuscript. Please add the link or repository where the code is uploaded.

Minor point: I know the authors present their plasticity rule as a phenomenological model rather than a mechanistic model, but if they have an idea of what "some continuous physiological signal represented by mu" might be, they might want to mention it in a sentence in the Discussion.

**Have the authors made all data and (if applicable) computational code underlying the findings in their manuscript fully available?**

Reviewer #1: Yes

Reviewer #2: Yes

PLOS authors have the option to publish the peer review history of their article (what does this mean?). If published, this will include your full peer review and any attached files.

Reviewer #1: No

Reviewer #2: No

Figure Files:

Data Requirements:

Reproducibility:

References:

---

## [Editor Report · Decision Letter 2]

25 Apr 2024

Dear Mr. Albesa Gonzalez,

We are pleased to inform you that your manuscript 'Learning with filopodia and spines: complementary strong and weak competition lead to specialized, graded, and protected receptive fields.' has been provisionally accepted for publication in PLOS Computational Biology.

Best regards,

Daniele Marinazzo

Section Editor

PLOS Computational Biology

Daniele Marinazzo

Section Editor

PLOS Computational Biology

---

## [Editor Report · Acceptance letter]

7 May 2024

PCOMPBIOL-D-23-01392R2 

Learning with filopodia and spines: complementary strong and weak competition lead to specialized, graded, and protected receptive fields.

Dear Dr Albesa-González,

I am pleased to inform you that your manuscript has been formally accepted for publication in PLOS Computational Biology. Your manuscript is now with our production department and you will be notified of the publication date in due course.

With kind regards,

Lilla Horvath
